# Inclusive, exclusive and hierarchical atlas of NFATc1⁺/PDGFR-α⁺ cells in dental and periodontal mesenchyme

Xue Yang[1,2†], Chuyi Han[1†], Changhao Yu[1,2], Bin Zhou[3], Ling Ye[1,2]*, Feifei Li[1,4]*, Fanyuan Yu[1,2]*

[1]State Key Laboratory of Oral Diseases & National Clinical Research Center for Oral Diseases, West China Hospital of Stomatology, Sichuan University, Chengdu, China; [2]Department of Endodontics, West China Hospital of Stomatology, Sichuan University, Chengdu, China; [3]Center for Excellence in Molecular Cell Science, University of Chinese Academy of Sciences, Shanghai, China; [4]Department of Pediatric Dentistry, West China Hospital of Stomatology, Sichuan University, Chengdu, China

**\*For correspondence:**
yeling@scu.edu.cn (LY);
feidepingfanshijie@163.com (FL);
fanyuan_yu@outlook.com (FY)

[†]These authors contributed equally to this work

## eLife Assessment

Utilizing transgenic lineage tracing techniques and tissue clearing-based advanced imaging and three-dimensional slices reconstruction, the authors comprehensively mapped the distribution atlas of NFATc1+ and PDGFR-α+ cells in dental and periodontal mesenchyme and tracked their in vivo fate trajectories. This **important** work extends our understanding of NFATc1+ and PDGFR-α+ cells in dental and periodontal mesenchyme homeostasis, and should provide impact on clinical application and investigation. The strength of this work is **compelling** in employing CRISPR/Cas9-mediated gene editing to generate two dual recombination systems, and mapped gNFATc1+ and PDGFR-α+ cells residing in dental and periodontal mesenchyme, their capacity for progeny cell generation, and their inclusive, exclusive and hierarchical relations in homeostasis, generating a spatiotemporal atlas of these skeletal stem cell population.

**Abstract** Platelet-derived growth factor receptor alpha (PDGFR-α) activity is crucial in the process of dental and periodontal mesenchyme regeneration facilitated by autologous platelet concentrates (APCs), such as platelet-rich fibrin (PRF), platelet-rich plasma (PRP) and concentrated growth factors (CGF), as well as by recombinant PDGF drugs. However, it is largely unclear about the physiological patterns and cellular fate determinations of PDGFR-α⁺ cells in the homeostasis maintaining of adult dental and periodontal mesenchyme. We previously identified NFATc1 expressing PDGFR-α⁺ cells as a subtype of skeletal stem cells (SSCs) in limb bone in mice, but their roles in dental and periodontal remain unexplored. To this end, in the present study we investigated the spatiotemporal atlas of NFATc1⁺ and PDGFR-α⁺ cells residing in dental and periodontal mesenchyme in mice, their capacity for progeny cell generation, and their inclusive, exclusive and hierarchical relations in homeostasis. We utilized CRISPR/Cas9-mediated gene editing to generate two dual recombination systems, which were Cre-*loxP* and Dre-*rox* combined intersectional and exclusive reporters respectively, to concurrently demonstrate the inclusive, exclusive, and hierarchical distributions of NFATc1⁺ and PDGFR-α⁺ cells and their lineage commitment. By employing the state-of-the-art transgenic lineage tracing techniques in cooperating with tissue clearing-based advanced imaging and three-dimensional slices reconstruction, we systematically mapped the distribution atlas of NFATc1⁺ and PDGFR-α⁺ cells in dental and periodontal mesenchyme and tracked their in vivo fate

trajectories in mice. Our findings extend current understanding of NFATc1⁺ and PDGFR-α⁺ cells in dental and periodontal mesenchyme homeostasis, and furthermore enhance our comprehension of their sustained therapeutic impact for future clinical investigations.

## Introduction

Platelet-derived growth factor (PDGF), first identified in platelets, intricately involved in the process of tissue regeneration facilitated by autologous platelet concentrates (APCs; *Andrae et al., 2008*; *Dereka et al., 2006*). The effects of PDGFs are mediated by two receptor tyrosine kinases, namely PDGFRα and PDGFRβ. The PDGFRα homodimer predominantly binds to the majority of PDGF isoforms (PDGF-AA, PDGF-AB, PDGF-BB, and PDGF-CC; *Heldin and Lennartsson, 2013*). In recent decades, APCs have gained widespread utilization in the field of tissue regenerative medicine, with notable generations including platelet-rich plasma (PRP), platelet-rich fibrin (PRF), and concentrated growth factors (CGF). Evidence-based medicine studies have demonstrated that these three APC techniques have been utilized in the field of periodontology, contributing to periodontal regenerative therapy, albeit with varying efficacy (*Mijiritsky et al., 2021*). Additionally, certain investigations suggest that PDGF-BB also exerted a role in promoting regeneration of both periodontal tissues and dental pulp (*Komatsu et al., 2022*; *Zhang et al., 2017*). Teeth, being highly mineralized organs, enclose a mineralized chamber housing the dental pulp mesenchyme, while the surrounding tissue comprises periodontal soft tissue—periodontal ligament (PDL; *Woelfel and Scheid, 1997*). Simply, teeth can be regarded as hard tissues enveloped by mesenchyme. While the dental pulp and periodontium fulfill distinct physiological roles, they also maintain interconnectedness to collectively sustain the physiological function of the tooth. Consequently, when pathological changes occur independently or concurrently in the dental pulp and periodontal tissues, they may mutually influence each other (*Ouchi and Nakagawa, 2020*). Therefore, it is imperative to elucidate the expression and biological mechanism of PDGFR-α⁺ cells in PDL and dental pulp tissues in order to comprehensively comprehend the physiological and therapeutic effects mentioned above.

PDGFR-α serves as a pivotal signaling molecule involved in mediating tooth development, morphogenesis, and physiological responses (*Chai et al., 1998*; *Guo et al., 2024*; *Xu et al., 2005*). Its potential role in periodontal formation has also been suggested (*Cui et al., 2021*). Within skeletal tissues, PDGFR-α has been identified as a heterogeneous population, with not all PDGFR-α⁺ cells exhibiting mesenchymal stem cell characteristics (*Morikawa et al., 2009*; *Uezumi et al., 2014*; *Kfoury and Scadden, 2015*; *Ning et al., 2022*). Certain studies suggest that subpopulations of PDGFR-α⁺ cells are considered 'true' stem cells. For instance, co-labeling of Nestin and PDGFR-α has been observed in bone marrow stem cells (BMSCs) derived from the neural crest (*Changmeng et al., 2023*), and the co-expression of PDGFR-α and Sca-1 in the non-hematopoietic compartment has identified a distinct cell population within the adult murine bone marrow (*Koide et al., 2007*). Our prior investigations have also revealed a population of NFATc1 and PDGFR-α double-positive skeletal stem cells (SSCs) within long bones (*Yu et al., 2022*). In long bones, NFATc1⁺ cells are considered SSCs (*Yu et al., 2022*). Furthermore, previous RNA-sequencing analyses have reported the expression of NFATc1 in jawbones and periodontal tissues (*Nassif et al., 2022*). However, this evidence was limited to RNA-level sequencing, lacking data support in vivo.

The conventional single-enzyme cutting system, represented by the Cre-*loxP* system primarily focuses on individual cells (*Buckingham and Meilhac, 2011*). The introduction of multiple DNA recombinases has revolutionized lineage tracing by enabling a layered and intersectional analysis that was previously unattainable (*Anastassiadis et al., 2009*; *Liu et al., 2020*). Notably, the utilization of multiple recombinase-mediated recombination includes systems such as intersectional reports (IR) and exclusive reporters (ER), which facilitate the visualization of two sets of orthogonal recombination or hierarchical relationships (*Liu et al., 2020*). In this study, considering significant tissue variations in dental pulp and PDL, traditional slicing methods may inadequately represent their expression patterns (*Yao et al., 2023*). To ensure accurate depiction of expression patterns in spatial dimensions, we employed two strategies: tissue transparency (TC) and three-dimensional (3D) slices reconstruction, guaranteeing comprehensive spatial imaging technology for investigating tissue structural architecture.

In summary, we utilized the advanced Cre-*loxP* and Dre-*rox* combined intersectional and exclusive reporters—IR and ER systems along with advanced imaging methods to comprehensively examine the intricate relationships, including inclusive, exclusive and hierarchical between two distinct cell populations within pulp and periodontal mesenchyme for the first time. Furthermore, we demonstrated their capacity for progeny cell generation and hierarchical positioning to depict their in vivo fate determination trajectories. Additionally, we assessed the advantages and limitations of TC and 3D slices reconstruction techniques. These findings not only furnish essential tools and references for future investigations into MSC lineage tracing within dental and periodontal tissues but also enhance our comprehension of biological research and therapeutic applications in regenerative medicine.

## Results

## Establishment of multiple genetic recombination systems and advanced imaging technology

To understand the cell fate, behaviors, and their spatiotemporal correlation of PDGFR-α$^+$ and NFATc1$^+$ cells in dental and periodontal mesenchyme, systematic genetic fate atlas were established by lineage tracing based on genetic DNA recombination technology. In conventional single recombinase-mediated genetic readouts, the Cre-*loxP* system is widely used for mammalian gene editing with Cre recombinase catalyzing recombination between *loxP* sites. This system, often activated by a ubiquitous promoter like *Rosa26* and *H11*, removes a transcription stop cassette post-recombination to allow for reporter gene expression (*Figure 1A*). However, the drawback of the conventional approach is that it is impossible to observe the genetic lineage and spatial relationship of two specific cell type at the same time, also, targeting two gene promoters in one cell population could be more precise than relying on a single promoter as commonly employed in the conventional reporter system. Given that, diverse dual recombinase–mediated genetic labeling systems have been developed to enhance the specificity and the number of cell types being labeled simultaneously. Cre-*loxP*, Flp-*frt*, Dre-*rox,* and Nigri-*nox* have been respectively used for designing dual system. Using reporter as an entry site can categorize these systems into three different types for multiple recombinase–mediated fate mapping studies, including intersectional reporters, exclusive reporters, and nested reporters (*Figure 1A–b and c*). In this work, the most advanced transgenic technology, dual recombinase–mediated genetic labeling systems, including intersectional reporters and exclusive reporters, were constructed to simultaneously show the inclusive, exclusive and hierarchical distribution of PDGFR-α$^+$/ NFATc1$^+$ cells in dental and periodontal mesenchyme. Specifically, multichromatic crossover intersectional reporters (MCIR) systems, which structured as *H11-CAG-LSL-ZsGreen-CAG-RSR-tdTomato* (LGRT), enabled the tracing of three cell types: Cre$^+$&Dre$^-$, Cre$^-$&Dre$^+$, and Cre$^+$&Dre$^+$, corresponding to the colors ZsGreen, tdTomato, and yellow, respectively. This technology guaranteed us to precisely understand the respective and inclusive distribution of Pdgfr-α$^+$, NFATc1$^+$ cells and their progeny cells. In addition, IR1 system is an exclusive reporter expressed as *CAG-loxP-rox-Stop-loxP-ZsGreen-Stop-rox-tdTomato* (*Figure 1A–c*). The first Cre-*loxP* recombination results in ZsGreen expression and the removal of a *rox* site, preventing subsequent Dre-*rox* recombination in the same cell. This means that once a recombination event occurs, the expression of the corresponding reporter gene eliminates a recognition site for the other recombination system, which could provide generous information of the hierarchical distribution of PDGFR-α$^+$ and NFATc1$^+$ cells in dental and periodontal mesenchyme.

Due to the peculiarity of dental/periodontal tissues, including pulp and periodontal of maxillary and mandibular first molars (small volume, complex structure and uneven spatial distribution of SSCs in mesenchyme), single-slice images based on traditional frozen section and confocal microscope imaging were insufficient to accurately presume the practical situation of the organization. Therefore, the advanced tissue deep clearance using SUMIC procedure and 3D reconstruction was exploited to observe the overall spatial distribution of PDGFR-α$^+$ and NFATc1$^+$ cells in dental and periodontal mesenchyme of transgenic mice with multiple DNA recombinases-based genetic lineage tracing system (*Figure 1C–a*). Simultaneously, as an irreplaceable imaging technology, traditional sectioning combined confocal imaging and procedure was also estimated (*Figure 1C–b*). These two 3D-reconstruction and imaging technologies complement each other to jointly address the spatial positioning and hierarchical relationships of PDGFR-α$^+$, NFATc1$^+$, and PDGFR-α$^+$ NFATc1$^+$ cells from multiple perspectives.

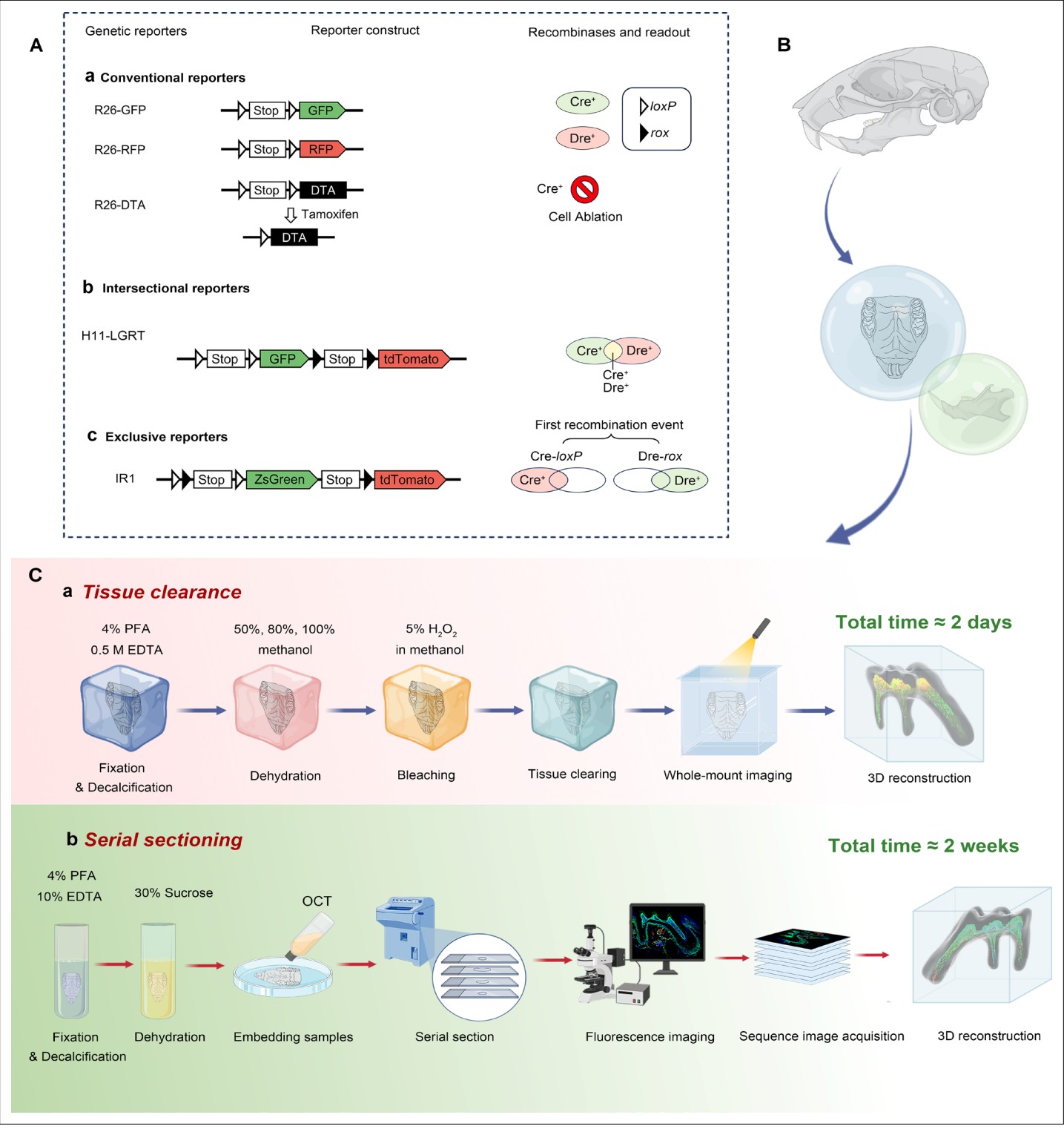

**Figure 1.** Advanced genetic recombination and imaging techniques for craniofacial tissue analysis in transgenic models. (**A**) Genetic recombination systems.The initial column lists various reporter alleles, while the subsequent second and third columns present the corresponding reporter constructs and the recombinases utilized. The final column provides the outcomes associated with the respective reporter alleles. Traditional reporter mechanisms comprise both single-color and multicolor systems (not shown in the figure), created through a singular recombination event. Advanced dual-reporter configurations encompass intersectional reporters, exclusive reporters, and nested reporters. (**B**) The craniofacial hard tissue, including maxilla and mandible of transgenic mice with multiple DNA recombinases-based genetic lineage tracing system, were obtained for advanced imaging technologies. (**C–a**) The tissue deep clearing procedure based on SUMIC, which needed to go through 4% PFA fixation, 0.5 M EDTA decalcification,

*Figure 1 continued on next page*

Figure 1 continued

different concentration of methanol dehydration, 5% $H_2O_2$ bleaching, ECI tissue clearing, and finally used the light-sheet system to directly image the whole tissue. (**C–b**) The traditional frozen section technology needs fixation, decalcification, dehydration and other steps, which takes about 2 weeks.

## Exploration of NFATc1⁺/PDGFR-α⁺ cells and their progeny cells in MCIR system

First, we established Cre-*loxP* and Dre-*rox* reporter lineage-tracking model of PDGFR-α and NFATc1 in adult mice to observe the spatial location of PDGFR-α⁺ and NFATc1⁺ cells in pulp and periodontal of maxilla & mandible M1. The pulse timing point we set was tamoxifen treated twice at day 1 and day 3, and then the mice were sacrifice to obtain maxilla and mandible on day 5 (*Figure 2A and B*).

## Dental and periodontal region of maxilla M1 and mandible M1

As one of the most advanced imaging technologies currently available, tissue clearing/imaging allows for direct observation of the spatial location and relationships of fluorescently labeled cells within the intact tissue. Therefore, according to the existing SUMIC tissue deep clearing (TC) methods (*Antila et al., 2024*; *Biswas et al., 2023*), we modified and improved a rapid and efficient procedure, which enable rapid single-cell resolution and quantitative panoptic 3D light-sheet imaging. This procedure not only rapidly clear maxillary hard tissue, but also directly penetrate the dentin to visualize the dental pulp. The total time of this procedure was only 2 days including tissue fixation and decalcification, this ultrafast clearing technique could minimize the impact on transgenic mice. As shown in *Figure 2—figure supplement 1A–B*, we recorded bright-field images of the maxilla before and after clearing, and our procedure achieved high transparency of the whole tissue. On this basis, whole-tissue imaging can be achieved, with the observation of different cell type distribution in spatial 3D structure. As such, dental pulp and periodontal ligament (PDL) region of M1 were representatively delineated for 3D reconstruction and analysis. *Figure 2* and *Figure 2—video 1* showed the contoured maxilla M1, including pulp and periodontal ligament (PDL) with virtual dentin shell (white). The buccal view, coronal view, and radicular view of pulp and PDL showed that PDGFR-α⁺ cells were equidistributional in the whole area of M1 pulp (including crown pulp and root pulp) and PDL area, almost constitute the whole dental and periodontal tissue, while NFATc1⁺ cells were almost exclusively distributed in crown pulp and scattered existence in PDL. Across the board, the distribution of NFATc1⁺ cells were significantly less than that of PDGFR-α⁺ cells in both pulp and PDL area, indicated that the expansive range of PDGFR-α⁺ cells which may contain NFATc1⁺ population. As mentioned above, studies have shown that with not all PDGFR-α⁺ cells within long bones exhibiting mesenchymal stem cell characteristics, and only certain specific subpopulations of PDGFR-α⁺ cells can be considered as 'real' stem cells. In addition, our previous study has evidenced NFATc1⁺PDGFR-α⁺ cells as skeletal stem cells (SSCs) within long bones, played an indispensable role in bone homeostasis and regeneration. As such, these phenomena we observed from the results of TC preliminarily indicate a similar situation in dental and periodontal tissue, and the biological role of NFATc1⁺PDGFR-α⁺ cells in homeostasis and regeneration situation is our major focus in the future research.

In addition, from the section of XZ axis after 3D reconstruction, NFATc1⁺ cells were mainly distributed in the alveolar bone area, including the surface edge of trabecular bone (*Figure 2—figure supplement 1C*), which was consistent with RNA-sequencing results in the previous study (*Nassif et al., 2022*). For the cross-sectional analysis of the 3D reconstructed data, the maximum cross-section of XZ axis which containing the whole pulp and PDL and the XY axis of the root pulp were selected (*Figure 2E*). However, the image is not ideal, which due to: (1) the editing efficiency DNA recombinase–mediated lineage-tracing system have limitations; (2) the less amount presence of NFATc1⁺ cells in the tissue guaranteed the week signals; (3) the as-describe TC method may cause poor penetration of the td-tomato fluorescence signals.

Therefore, in order to obtain high-resolution image to understand more detailed information, we still need to rely on the traditional serial section technology (*Figure 3A*). A total of 121 slices were collected in this maxilla of *Pdgfra^CreER^×Nfatc1^DreER^×* LGRT mice sample, from the jaw root PDL just appeared, to the buccal root PDL completely disappeared (10 μm/slice; *Figure 3—figure supplement 1*). As shown in *Figure 3B*, *Figure 3—figure supplement 2*, although time consuming, the images obtained by confocal imaging can clearly obtain the distribution of PDGFR-α⁺ and NFATc1⁺

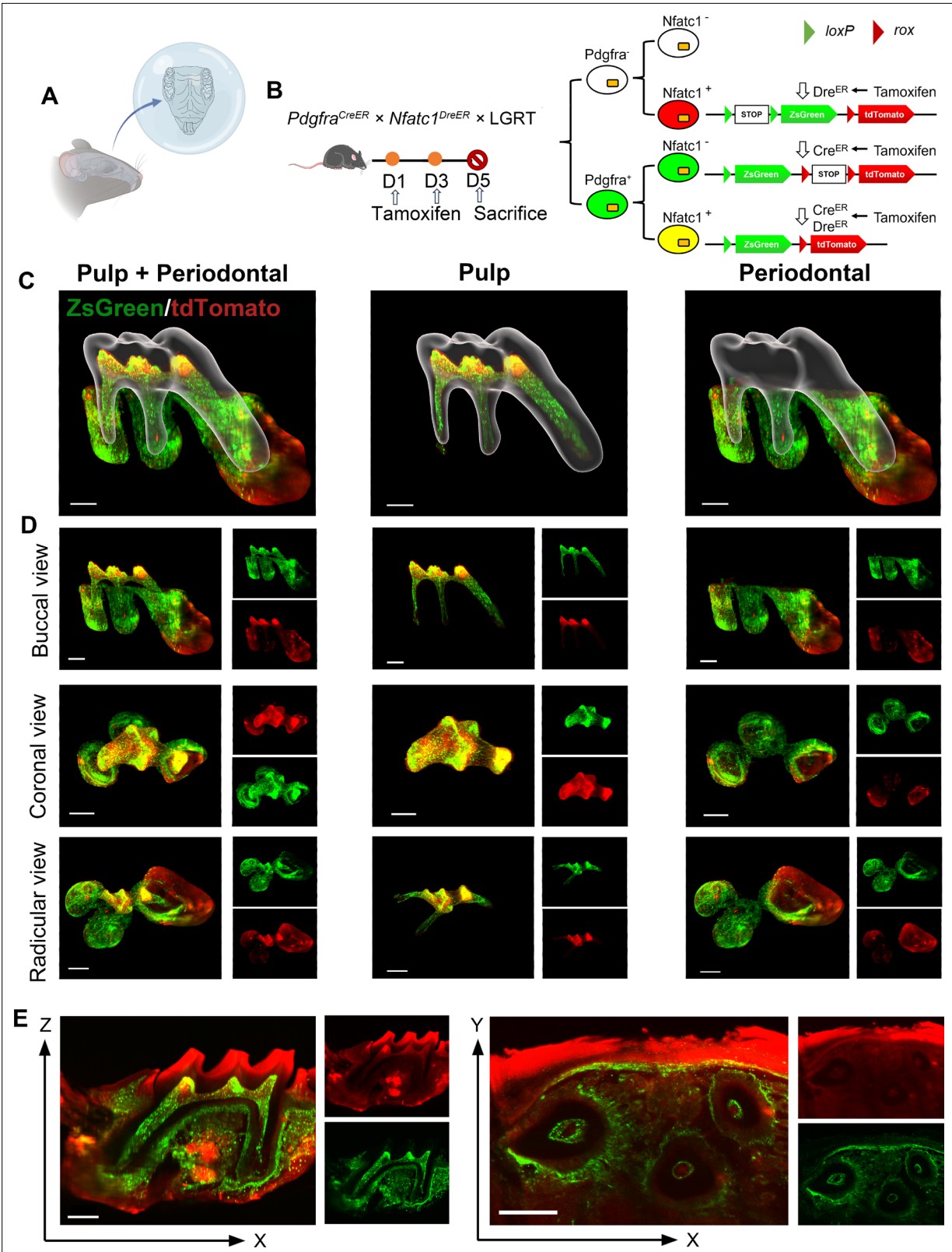

**Figure 2.** The observation of PDGFR-α⁺ and NFATc1⁺ cells in maxilla M1 of *Pdgfra^CreER^ × Nfatc1^DreER^* × LGRT mice (pulse) by whole-mount and high-speed imaging. The whole-mount and high-speed imaging of (**A**) mouse molar with a tiling light-sheet microscope. (**B**) Schematic illustration of lineaging tracing in *Pdgfra^CreER^×Nfatc1^DreER^×* LGRT mice. The mice were administrated with tamoxifen at D1 and D3, and sacrificed at D5. (**C**) The contoured M1 of maxilla, including pulp and PDL with virtual dentin shell (white) in buccal view (scale bar = 300 μm). (**D**) Image stack was displayed in

*Figure 2 continued on next page*

*Figure 2 continued*

buccal view, coronal view, and radicular view of pulp and PDL, respectively (scale bar = 300 μm). (**E**) An optical slice was acquired on the X-Z (scale bar = 300 μm) and X-Y direction (scale bar = 400 μm) to display the pulp and PDL.

The online version of this article includes the following video and figure supplement(s) for figure 2:

**Figure supplement 1.** SUMIC procedure achieves whole-mount transparency of mice maxilla and enables imaging of dental, periodontal and alveolar bone.

**Figure 2—video 1.** Panoptic multicolor imaging of PDGFR-α⁺ cells(green) & NFATc1⁺ cells(red) in the pulp and PDL area of maxilla M1 of *Pdgfra*^CreER^×*Nfatc1*^DreER^× LGRT mice (pulse), the whole-tissue imaging was achieved through TC procedure, related to *Figure 2C*.
https://elifesciences.org/articles/100173/figures#fig2video1

cells from the single cell level. However, it is difficult to determine the presence of PDGFR-α⁺&N-FATc1⁺ cells spatiotemporally. So, we also used Imaris to 3D-reconstruct these 121 images. *Figure 3C* and *Figure 3—video 1* showed the contoured M1 of maxilla, including pulp and PDL with virtual dentin shell (white). *Figure 3D* was the buccal view, coronal view, and radicular view of pulp and PDL, respectively. Different from the whole imaging obtained by TC and light-sheet, the td-tomato signal reconstructed by traditional serial section-based confocal imaging method was very conspicuous, even in root pulp, which were hardly obtained by TC imaging (*Figure 3—figure supplement 3*). Same as TC-based reconstructed results, PDGFR-α⁺ cells almost constituted the whole structure of pulp and PDL, with NFATc1⁺ cells as subpopulation. However, due to the stratification of slices, the sample integrity was poor, leading to discontinuities in the z-axis. (*Figure 3—figure supplement 4*), which may render some biases in the results. To enhance the comprehensive and accurate display of the reconstruction results and to mitigate the potential errors that may arise from relying on single reconstruction method, we employed an alternative 3D reconstruction method—DICOM-3D (*Popescu et al., 2021*). This method is based on sequential 2D DICOM images and utilizes 3D reconstruction and visualization technology to generate a stereoscopic 3D image with intuitive effects, which was a comparatively straightforward and highly efficient approach. We transformed the serial IF images into DICOM format and subsequently reconstruct it, and the same conclusion can be drawn, namely, PDGFR-α⁺ cells almost constituted the whole structure of pulp and PDL, with NFATc1⁺ cells as subpopulation (*Figure 3—figure supplement 5*).

The previous sequencing analyses have reported the expression of NFATc1 in mandible and periodontal tissues (*Nassif et al., 2022*). However, this evidence is limited to RNA-level sequencing, lacking in vivo data support. Therefore, with assistance of multiple genetic recombination systems, we next examined the expression pattern of NFATc1 in dental and PDL region of mandible M1, simultaneously observed the distribution atlas of PDGFR-α⁺ cells. As shown in *Figure 4A*, the pulse timing points were same as above. *Figure 4C* and *Figure 4—video 1* showed the contoured mandible M1, including pulp and PDL with virtual dentin shell (white) in the buccal view, coronal view, and radicular view of pulp and PDL, respectively. According to the 3D-reconstruction results of 88 consecutive slices, it can be concluded that PDGFR-α⁺ cells almost constituted the whole structure of dental pulp and periodontal ligament, while NFATc1⁺ cells most distributed in the pulp angle in the dental pulp tissue, and also scattered distributed in PDL region (*Figure 4C,D*, *Figure 4—figure supplement 1*). As shown in *Figure 4C*, *Figure 4—figure supplement 2*, the images obtained by confocal imaging can clearly obtain the distribution of PDGFR-α⁺ and NFATc1⁺ cells from the single cell level.

To further understand the presence of PDGFR-α⁺ cells and NFATc1⁺ cells and their progeny cells after long-term tracking, we set a tracing timing points (*Figure 5A*). The results showed that nearly the whole pulp and PDL were composed of PDGFR-α⁺ cells, and NFATc1⁺ cells were also scattered present in the root pulp and PDL region of maxilla M1 (*Figure 5B–D*, *Figure 5—video 1*). Statistical data quantitatively showed that after 11 days of tracing, the number of PDGFR-α⁺ cells, NFATc1⁺ cells, PDGFR-α⁺NFATc1⁺ cells, and their progeny cells increased in both dental pulp and PDL (*Figure 5—figure supplement 1*). Specifically, the number of NFATc1⁺ cells in PDL area after tracing 11 days was 2.19-fold that of pulse NFATc1⁺ cells. Further, we also conducted cross-sectional analysis of the 3D reconstructed data, and selected the maximum cross-section of XZ axis which containing the whole pulp and PDL and the XY axis of the root pulp, respectively (*Figure 5D*D). The results were similar to *Figure 2E*. From the cross section, only the Zs-Green fluorescence signal of PDGFR-α⁺ cells can be observed and analyze while the td-tomato signal was illegible in target area, which was unable to distinguish from background interference signal. In the process of sample preparation, the soft tissue

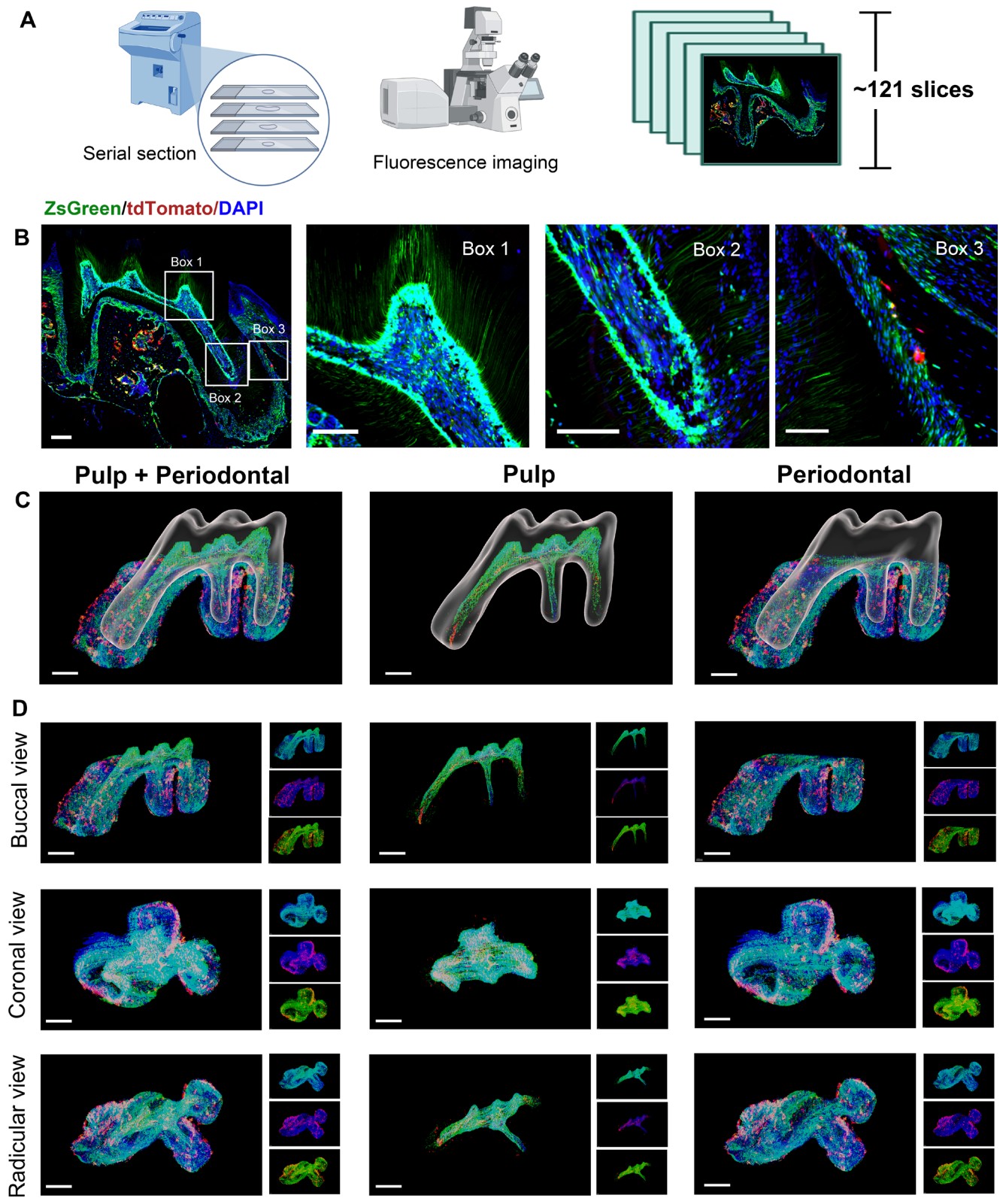

**Figure 3.** Using traditional serial section for imaging of maxilla of *Pdgfra^CreER^×Nfatc1^DreER^*× LGRT mice (pulse). (**A**) Operation process of frozen section. Using this procedure, a total of 121 slices were collected in this sample. (**B**) Representative images acquired by confocal microscopy (scale bar = 200 μm). Box 1: coronal pulp, Box 2: root pulp, Box 3: PDL, scale bar = 100 μm. (**C**) Maxilla M1 after 3D reconstruction by imaris, including pulp and PDL

*Figure 3 continued on next page*

*Figure 3 continued*

with virtual dentin shell (white) in buccal view (scale bar = 300 μm). (**D**) Image stack was displayed in buccal view, coronal view, and radicular view of pulp and PDL, respectively (scale bar = 300 μm).

The online version of this article includes the following video and figure supplement(s) for figure 3:

**Figure supplement 1.** The total 121 slices of maxilla M1 of *Pdgfra^CreER^×Nfatc1^DreER^×* LGRT mice sample (pulse).

**Figure supplement 2.** Representative images of coronal pulp, root pulp, and PDL of maxilla M1 of *Pdgfra^CreER^×Nfatc1^DreER^×* LGRT mice sample (pulse).

**Figure supplement 3.** The tdTomato signal in pulp reconstructed by traditional serial section-based confocal imaging method (scale bar = 300 μm).

**Figure supplement 4.** The discontinuities in the z-axis due to stratification of slices.

**Figure supplement 5.** 3D reconstruction of maxilla M1 of *Pdgfra^CreER^×Nfatc1^DreER^×* LGRT mice (pulse) by DICOM-3D; in PDL: ZsGreen⁺ cells in green, tdTomato⁺ cells in rose red; in pulp: ZsGreen⁺ cells in purple, tdTomato⁺ cells in blue.

**Figure 3—video 1.** Panoptic multicolor imaging of PDGFR-α⁺ cells & NFATc1⁺ cells in the pulp and PDL area of maxilla M1 of *Pdgfra^CreER^×Nfatc1^DreER^×* LGRT mice (pulse), the whole-tissue imaging was reconstructed from serial sections, related to *Figure 3C*.

https://elifesciences.org/articles/100173/figures#fig3video1

on the surface needs to be removed as much as possible before fixation, otherwise a similar situation will occur. However, for periosteum and other membranous structures that needed to be studied, the method of TC seems not suitable.

The tracing samples were also serially sliced and 3D reconstructed (*Figures 6 and 7*). A total of 117 slices were collected in maxilla of tracing sample (*Figure 6—figure supplement 1*) and a total of 120 slices were collected in mandible (*Figure 7—figure supplement 1*) from the jaw root PDL just appeared, to the buccal root PDL completely disappeared (10 μm/slice). As shown in *Figure 6B*, *Figure 4—figure supplement 2* and *Figure 7B*, *Figure 7—figure supplement 2*, the images obtained by confocal imaging can clearly obtain the distribution of PDGFR-α⁺ and NFATc1⁺ cells from the single cell level. Consistent with the quantification of TC-based imaging results (*Figure 5—figure supplement 1*), the number of PDGFR-α⁺ cells and NFATc1⁺ cells were significantly higher than that in pulse group, which indicated that there were a large number of PDGFR-α⁺ & NFATc1⁺ cells and their progeny cells in crown pulp, root pulp and PDL after 11 days of tracing. In order to determine the existence of PDGFR-α⁺NFATc1⁺ cells spatiotemporally, we also used software to 3D- reconstruct these 117 images. *Figure 6C–D* and *Figure 6—video 1* showed the contoured M1 of maxilla, including pulp and PDL, at buccal view, coronal view, and radicular view. *Figure 7C–D* and *Figure 7—video 1* showed the contoured mandible M1. Consistent with TC imaging and following quantitative results (*Figure 5—figure supplement 1*), the number of PDGFR-α⁺NFATc1⁺ cells increased significantly in both dental pulp and periodontal ligament, similar to the phenomenon previously observed in long bone, which further encouraged us to follow up with this cell population, explore whether it is a specific mesenchymal stem cell population in periodontal tissue, and explore the indispensable role in periodontal homeostasis and regeneration. Worth notably, different from the whole imaging obtained by TC and light-sheet, the tdTomato signal reconstructed by traditional serial section-based confocal imaging method was very conspicuous, no matter in coronal pulp, root pulp, or PDL (*Figure 6—figure supplement 3*). We hypothesize that the current light-sheet systems for intact tissue-imaging have inherent limitations in capturing tdTomato signals, which become more evident in tissues with inherently low fluorescence strengths (in this work, due to the limitations of editing efficiency in DNA recombinase mediated lineage-tracing system, which guaranteed weaker tdTomato signal compared to ZsGreen). In contrast, traditional confocal imaging techniques do not encounter such issues.

However, due to the stratification of slices, the sample integrity was poor, leading to discontinuities in the z-axis. (*Figure 6—figure supplement 4*), which may render some biases in the results. We also utilized 3D reconstruction of mandible M1 by DICOM-3D (*Figure 6—figure supplement 5*). After 11 days tracing, the number of PDGFR-α⁺ & NFATc1⁺ cells and PDGFR-α⁺NFATc1⁺ cells increased significantly (*Figure 7*), which further supported our above conjecture that NFATc1⁺ cells were a subset of PDGFR-α⁺ cells, and PDGFR-α⁺NFATc1⁺ cells may contribute to mesenchymal stem cell population in dental and PDL regions.

## Cranium and cranial sutures region

As a part of craniomaxillofacial hard tissue, we also intended to explore whether the presence of NFATc1⁺ and PDGFR-α⁺ cells in cranial bone tissue/suture is different from dental and periodontal

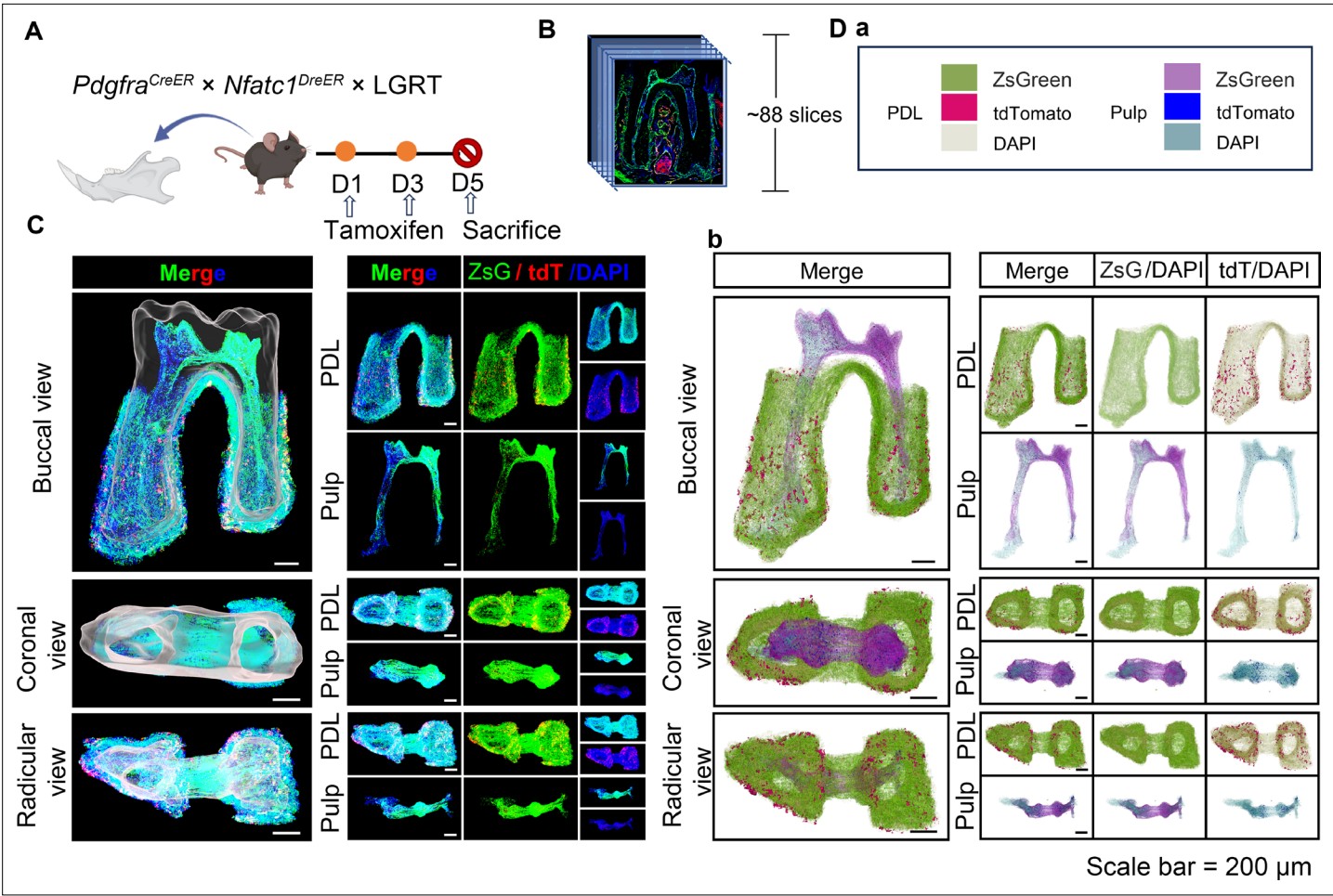

**Figure 4.** Using traditional serial section for imaging of mandible of *Pdgfra^{CreER}×Nfatc1^{DreER}×* LGRT mice (pulse). (**A**) Schematic illustration of lineaging tracing in *Pdgfra^{CreER}×Nfatc1^{DreER}×* LGRT mice. (**B**) Operation process of frozen section. Using this procedure, a total of 88 slices were collected in this sample. (**C**) Mandible M1 after 3D reconstruction by Imaris, including pulp and PDL with virtual dentin shell (white) in buccal view, coronal view and radicular view (scale bar = 200 μm). (**D**) 3D reconstruction of mandible M1 by DICOM-3D; in PDL, ZsGreen⁺ cells in green, tdTomato⁺ cells in rose red; in pulp, ZsGreen⁺ cells in purple, tdTomato⁺ cells in blue. The image stack was also displayed in buccal view, coronal view, and radicular view of pulp and PDL, respectively (**D-b**), scale bar = 200 μm. (**D-a**): The legend of (**D-b**).

The online version of this article includes the following video and figure supplement(s) for figure 4:

**Figure supplement 1.** The total 88 slices of mandible M1 of *Pdgfra^{CreER}×Nfatc1^{DreER}×* LGRT mice sample (pulse).

**Figure supplement 2.** Representative images of coronal pulp (A) and PDL (B) acquired by confocal microscope of mandible M1 of *Pdgfra^{CreER}×Nfatc1^{DreER}×* LGRT mice sample (pulse).

**Figure 4—video 1.** Panoptic multicolor imaging of PDGFR-α⁺ cells & NFATc1⁺ cells in the pulp and PDL area of mandible M1 of *Pdgfra^{CreER}×Nfatc1^{DreER}×* LGRT mice (pulse), the whole-tissue imaging was reconstructed from serial sections, related to *Figure 4C*.
https://elifesciences.org/articles/100173/figures#fig4video1

tissue (our previous study has identified the presence of NFATc1⁺ cells in the cranium by single-cell sequencing [*Yu et al., 2022*]), therefore, we also obtained the cranium of *Pdgfra^{CreER}×Nfatc1^{DreER}×* LGRT mice (pulse) for advanced TC imaging to familiarize NFATc1⁺/PDGFR-α⁺ cells in cranium and cranial sutures region. As shown in *Video 1*; *Video 2*; *Video 3*, PDGFR-α⁺ cells were widely distributed, including cranial sutures and bone interstitium, while NFATc1⁺ cells are mainly distributed in cranial sutures, partially coincided with the spatial location of PDGFR-α⁺ cells. The PDGFR-α⁺NFATc1⁺ cells located in the cranial suture were likely to participate in the physiological function of the cranial suture as a mesenchymal stem cell population.

The improved TC technology and following 3D imaging can deepen the understanding of specific cell populations in craniomaxillofacial tissues to a certain extent, and also lay a foundation for our

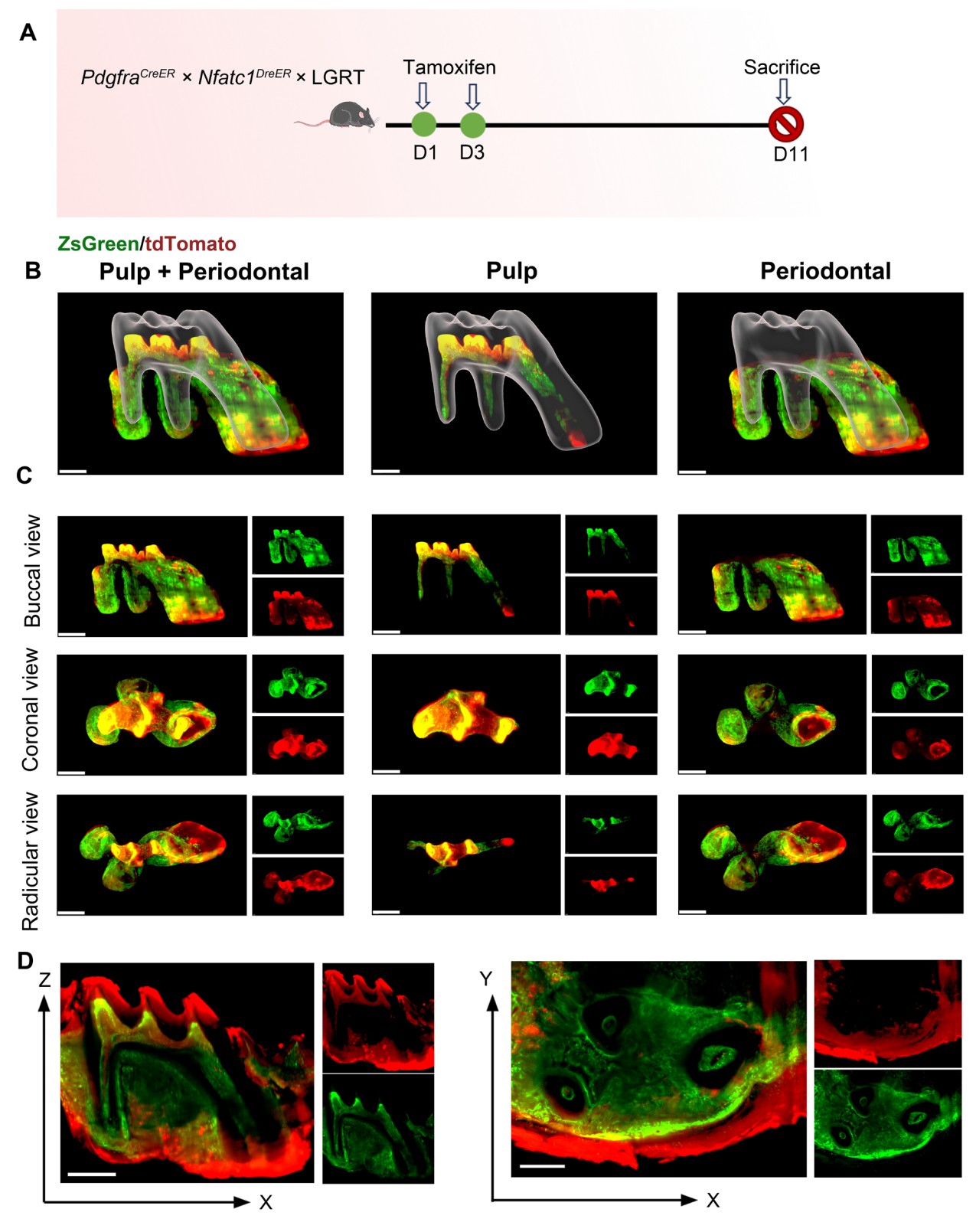

**Figure 5.** The observation of lineage tracing of PDGFR-$\alpha^+$ and NFATc1$^+$ cells in M1 by whole-mount and high-speed imaging. (**A**) The flowchart of tracing. The mice were administrated with tamoxifen at D1 and D3, and sacrificed at D11. (**B**) The 3D images of contoured M1 of maxilla, including pulp and PDL with virtual dentin shell (white) in buccal view (scale bar = 300 µm). (**D**) Image stack was displayed in buccal view, coronal view, and radicular

*Figure 5 continued on next page*

Figure 5 continued

view of pulp and PDL, respectively. (**E**) An optical slice was acquired on the X-Z (scale bar = 400 µm) and X-Y direction to display the pulp and PDL (scale bar = 300 µm).

The online version of this article includes the following video, source data, and figure supplement(s) for figure 5:

**Figure supplement 1.** The number of PDGFR-$\alpha^+$ cells, NFATc1$^+$ cells, and PDGFR-$\alpha^+$&NFATc1$^+$ cells in pulp and PDL, respectively.

**Figure supplement 1—source data 1.** The tables present the source data for the number of PDGFR-$\alpha^+$ cells, NFATc1$^+$ cells, and PDGFR-$\alpha^+$&NFATc1$^+$ cells in pulp and PDL, respectively.

**Figure 5—video 1.** Panoptic multicolor imaging of ZsGreen$^+$ cells (green) & tdTomato$^+$ cells (red) in the pulp and PDL area of maxilla M1 of *Pdgfra*$^{CreER}$×*Nfatc1*$^{DreER}$× LGRT mice (tracing 11 days), the whole-tissue imaging was achieved through TC procedure, related to **Figure 5B**.
https://elifesciences.org/articles/100173/figures#fig5video1

future researches. However, it seemed not suitable for the study of periosteum, due to the strong auto-fluorescence under the light film system of soft tissues, which will affect the observation of target cells.

## The hierarchical distribution of NFATc1$^+$/PDGFR-$\alpha^+$ cells in IR1 lineage tracing system

From the above MCIR system, the wider range of PDGFR-$\alpha^+$ cells than NFATc1$^+$ cells were observed, the wider range of PDGFR-$\alpha^+$ cells than NFATc1$^+$ cells were observed, we speculate that there may be a hierarchical relationship between the two. To further explore the relationship between PDGFR-$\alpha^+$ and NFATc1$^+$ cells in pulp and PDL and their hierarchical presence, respectively, we established the IR1 lineage tracing model in adult mice to assess the existence of these cells in the dental pulp and PDL. The structure of IR1 is *CAG-loxP-rox-Stop-loxP-ZsGreen-Stop-rox-tdTomato*, and the first Cre-*loxP* recombination would result in ZsGreen expression that removes a *rox* site, preventing the subsequent Dre-*rox* recombination in the same cell. The genotypes used were *Pdgfra*$^{CreER}$× IR1, *Nfatc1*$^{DreER}$× IR1 and *Pdgfra*$^{CreER}$×*Nfatc1*$^{DreER}$× IR1 (**Figure 8A–B**) in which NFATc1$^+$ cells expressed tdTomato and PDGFR-$\alpha^+$ cells were marked by ZsGreen. Tracing timing was set based on preliminary experimental outcomes. To comprehensively and accurately observe the distribution of NFATc1$^+$ and PDGFR-$\alpha^+$ cells in the pulp and PDL, we employed 3D reconstruction technology based on serial sections (**Figure 8C–E**, **Figure 8—videos 1–3** and **Figure 8—figure supplements 1–3**, and **Figure 8—figure supplement 5**). As shown in **Figure 8C-E**, **Figure 8—figure supplement 4**, the images obtained by confocal imaging can clearly obtain the distribution of ZsGreen$^+$ and tdTomato$^+$ cells from the single cell level. In *Nfatc1*$^{DreER}$× IR1 group, most tdTomato$^+$ cells distributed in PDL region, while only sparse and punctate distribution in dental pulp, mostly in coronal pulp. These results were consistent with *Pdgfra*$^{CreER}$×*Nfatc1*$^{DreER}$× LGRT mice. Moreover, since the recombinase recognition sites are interleaved (*loxP–rox–loxP–rox*), recombination by one system will naturally remove a recognition site of the other system, rendering its reporter gene inactive for further recombination. The results showed no tdTomato$^+$ cells or ZsGreen$^+$ cells were detected in the *Pdgfra*$^{CreER}$× IR1 or *Nfatc1*$^{DreER}$× IR1 group respectively demonstrating the feasibility and accuracy of the IR1 system. In the *Pdgfra*$^{CreER}$×*Nfatc1*$^{DreER}$× IR1 group, the distribution of Nfact1$^+$ cells in PDL were significantly fewer in number than in the *Nfatc1*$^{DreER}$× IR1 group, while no significant difference in pulp area. In addition, PDGFR-$\alpha^+$ cells were noted in both the dental pulp and PDL, with a wider distribution than NFATc1$^+$ cells, predominantly in the odontoblastic layer of the pulp and also present in the pulp core. In the PDL, they were distributed from the gingival direction down to the root apex. These phenomena illustrated that PDGFR-$\alpha^+$ cells were the precursor cell of NFATc1$^+$ cells in PDL region. By contrast, there was almost no difference in the number of NFATc1$^+$ cells between *Nfatc1*$^{DreER}$× IR1 group and *Pdgfra*$^{CreER}$×*Nfatc1*$^{DreER}$× IR1 group in dental pulp tissue, which was insufficient to certificate the hierarchical relationship of PDGFR-$\alpha^+$ and NFATc1$^+$ cells in pulp area.

## Identification of PDGFR-$\alpha^+$ and NFATc1$^+$ population in pulp and PDL

### Ablation of PDGFR-a$^+$ cells disrupt the morphology of dental pulp and periodontal mesenchymal tissues

The results from the aforementioned lineage tracing experiments showed that PDGFR-$\alpha^+$ cells constitute a significant component of both dental pulp and periodontal tissues. Additionally, the hierarchical

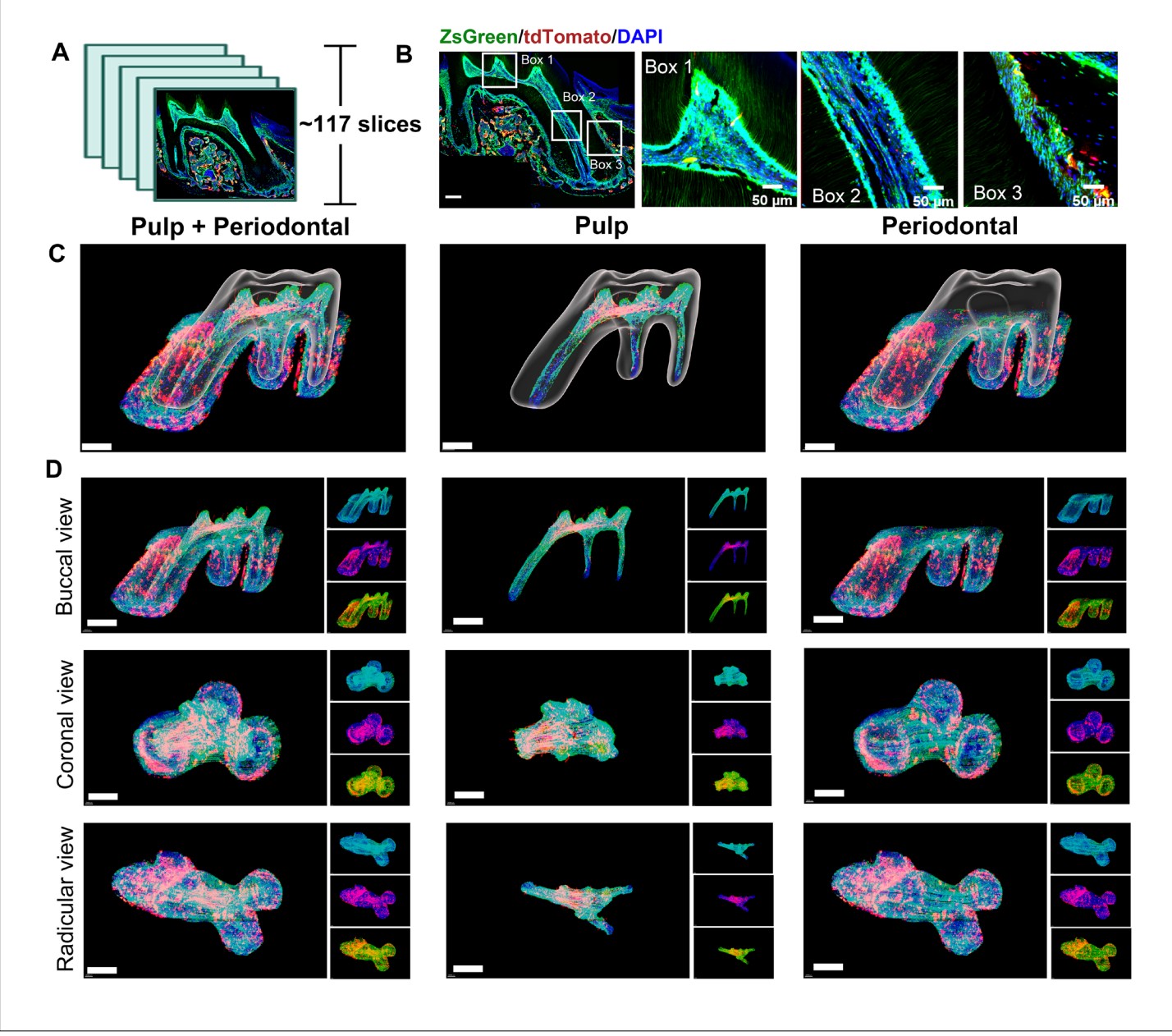

**Figure 6.** Using traditional serial section for imaging of maxilla of *Pdgfra*<sup>CreER</sup>×*Nfatc1*<sup>DreER</sup>× LGRT mice (tracing 11 days). (**A**) Using this procedure, a total of 117 slices were collected in this sample. (**B**) Representative images acquired by confocal microscopy (scale bar = 200 μm). Box 1: coronal pulp; Box 2: root pulp; Box 3: PDL (scale bar = 50 μm). (**C**) Maxilla M1 after 3D reconstruction by imaris, including pulp and PDL with virtual dentin shell (white) in buccal view (scale bar = 300 μm). (**D**) Image stack was displayed in buccal view, coronal view, and radicular view of pulp and PDL, respectively (scale bar = 300 μm).

The online version of this article includes the following video and figure supplement(s) for figure 6:

**Figure supplement 1.** The total 117 slices of maxilla M1 of *Pdgfra*<sup>CreER</sup>×*Nfatc1*<sup>DreER</sup>× LGRT mice sample (tracing).

**Figure supplement 2.** Representative images of coronal pulp, root pulp, and PDL of maxilla M1 of *Pdgfra*<sup>CreER</sup>×*Nfatc1*<sup>DreER</sup>× LGRT mice sample (tracing).

**Figure supplement 3.** The tdTomato signal in pulp of maxilla M1 of *Pdgfra*<sup>CreER</sup>×*Nfatc1*<sup>DreER</sup>× LGRT mice sample (tracing) reconstructed by traditional serial section-based confocal imaging method (scale bar = 300 μm).

**Figure supplement 4.** The discontinuities in the z-axis due to stratification of slices.

**Figure supplement 5.** 3D reconstruction of maxilla M1 of *Pdgfra*<sup>CreER</sup>×*Nfatc1*<sup>DreER</sup>× LGRT mice (tracing) by DICOM-3D; PDL: ZsGreen⁺ cells in green, tdTomato⁺ cells in rose red; pulp: ZsGreen⁺ cells in purple, tdTomato⁺ cells in blue.

**Figure 6—video 1.** Panoptic multicolor imaging of ZsGreen⁺ cells & tdTomato⁺ cells in the pulp and PDL area of maxilla M1 of *Pdgfra*<sup>CreER</sup>×*Nfatc1*<sup>DreER</sup>×

*Figure 6 continued on next page*

relationship experiments revealed that a portion of NFATc1[+] cells in the periodontal ligament derives from PDGFR-α[+] progenitor cells. Therefore, investigating the role of PDGFR-α[+] cells in dental pulp and periodontal tissues has become more urgent. To explore the role of PDGFR-α[+] cells in dental pulp and PDL tissues, we performed the cell ablation assay using *Pdgfra*[CreER]×*DTA* mice with tamoxifen administered for 12 weeks (*Figure 9A*). The *Pdgfra*[CreER]×*DTA* mice strain harbors a *loxP*-flanked termination cassette within the ubiquitous ROSA26 locus, controlling the expression of DTA. Tamoxifen administration was employed for in vivo toxin-mediated cell ablation (*Figure 9B*). Through H&E staining and

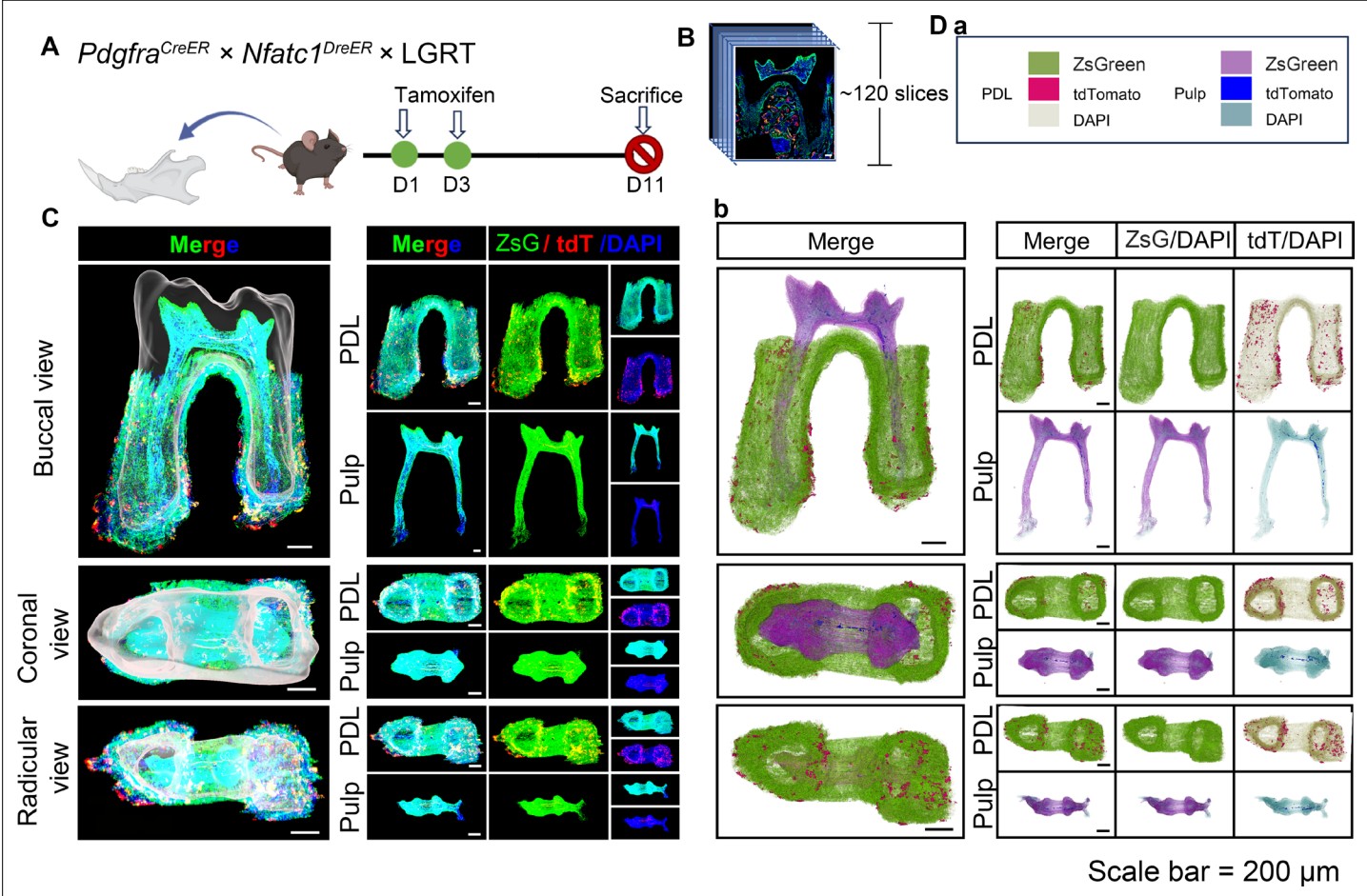

**Figure 7.** MCIR tracing distinct cell populations simultaneously in the pulp and PDL. (**A**) Schematic illustration of pulse in *Pdgfra*[CreER]×*Nfatc1*[DreER]× LGRT mice. (**B**) Operation process of frozen section. Using this procedure, a total of 120 slices were collected in this sample. (**C**) 3D reconstruction of mandible M1 using Imaris, including pulp and PDL with virtual dentin shell (white) (scale bar = 200 µm). The image stack was displayed in buccal view, coronal view, and radicular view of pulp and PDL, respectively. (**D**) 3D reconstruction of mandible M1 using DICOM-3D, in which the distribution of NFATc1[+] cells in pulp and PDL tissues can be more precisely perceived (**D-b**), scale bar = 200 µm. (**D-a**): The legend of (**D-b**).

The online version of this article includes the following video and figure supplement(s) for figure 7:

**Figure supplement 1.** All consecutive slices (a total of 120 slices) for imaging of mandible M1 of *Pdgfra*[CreER]×*Nfatc1*[DreER]× LGRT mice (tracing 11 days).

**Figure supplement 2.** Representative images of coronal pulp (**A**) and PDL (**B**) acquired by confocal microscope of mandible M1 of *Pdgfra*[CreER]×*Nfatc1*[DreER]× LGRT mice sample (tracing).

**Figure 7—video 1.** Panoptic multicolor imaging of ZsGreen[+] cells (green) & tdTomato[+] cells (red) in the pulp and PDL area mandible M1 of *Pdgfra*[CreER]×*Nfatc1*[DreER]× LGRT mice (tracing 11 days), the whole-tissue imaging was reconstructed from serial sections, related to *Figure 7C*.
https://elifesciences.org/articles/100173/figures#fig7video1

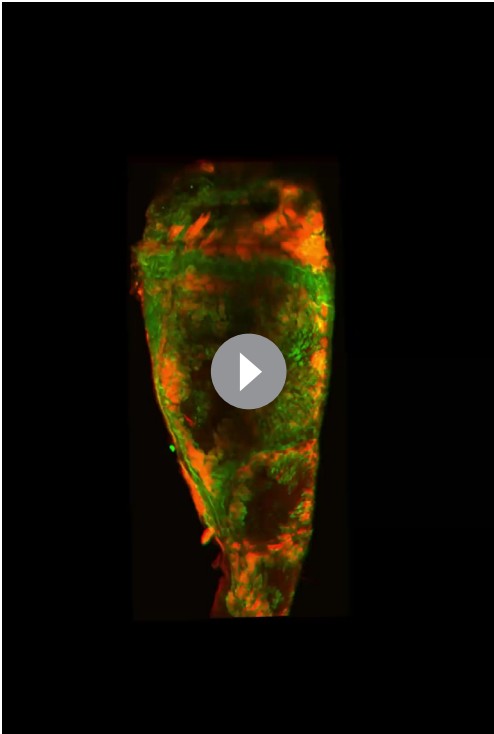

**Video 1.** Panoptic multicolor imaging of PDGFR-α⁺ cells (green) & NFATc1⁺ cells (red) in cranium of *Pdgfra*^CreER^×*Nfatc1*^DreER^× LGRT mice (pulse), the whole-tissue imaging was achieved through TC procedure.
https://elifesciences.org/articles/100173/figures#video1

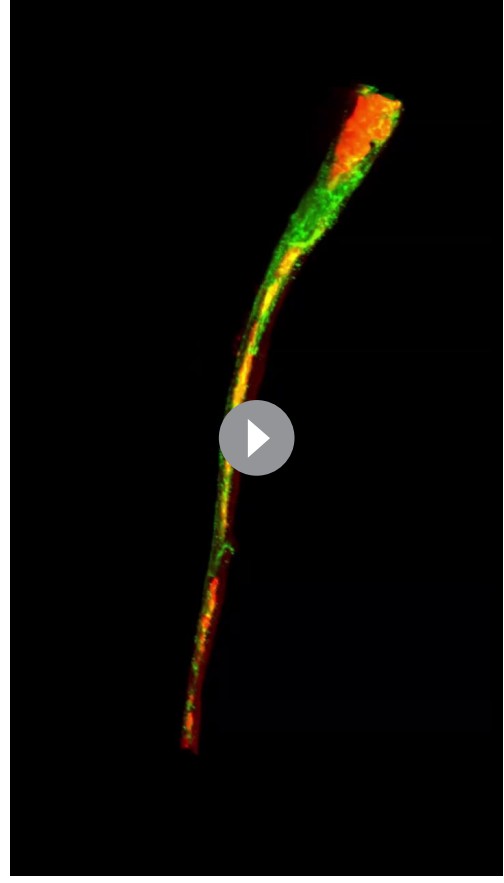

**Video 2.** Panoptic multicolor imaging of PDGFR-α⁺ cells (green) and NFATc1⁺ cells (red) in cranial sagittal suture of *Pdgfra*^CreER^×*Nfatc1*^DreER^× LGRT mice (pulse), the whole-tissue imaging was achieved through TC procedure.
https://elifesciences.org/articles/100173/figures#video2

Masson trichrome staining, we observed that in the control group, the odontoblast cell layer adjacent to the dentin consisted of densely arranged columnar cells both in the coronal or radicular pulp. However, this layer of odontoblasts was disrupted and nearly disappeared after ablation. Additionally, ablation of PDGFR-α⁺ cells in adult DTA mice also showed significant shrinkage in the central area of dental pulp tissue, specifically within the region of the pulp core, along with a substantial reduction in cell numbers (*Figure 9C–D*). H&E staining confirmed a decrease in periodontal composition following PDGFR-α⁺ cell clearance (*Figure 9E*). Furthermore, Masson's staining results revealed that the diminished portion primarily comprised periodontal fibrous tissue (*Figure 9F*). Therefore, it can be inferred that PDGFR-α⁺ cells predominantly constitute the interstitial components of both the periodontium and dental pulp. PDGFR-α⁺ cells within the dental pulp play a crucial role in the formation of the odontoblast cell layer and contribute significantly to most of the core tissue, while PDGFR-α is primarily involved in shaping periodontal components, particularly fibrous tissue.

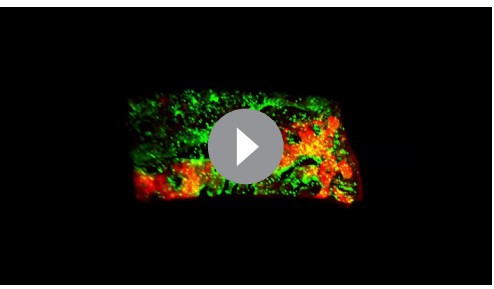

**Video 3.** Panoptic multicolor imaging of PDGFR-α⁺ cells (green) and NFATc1⁺ cells (red) in cranial coronal suture of *Pdgfra*^CreER^×*Nfatc1*^DreER^× LGRT mice (pulse), the whole-tissue imaging was achieved through TC procedure.
https://elifesciences.org/articles/100173/figures#video3

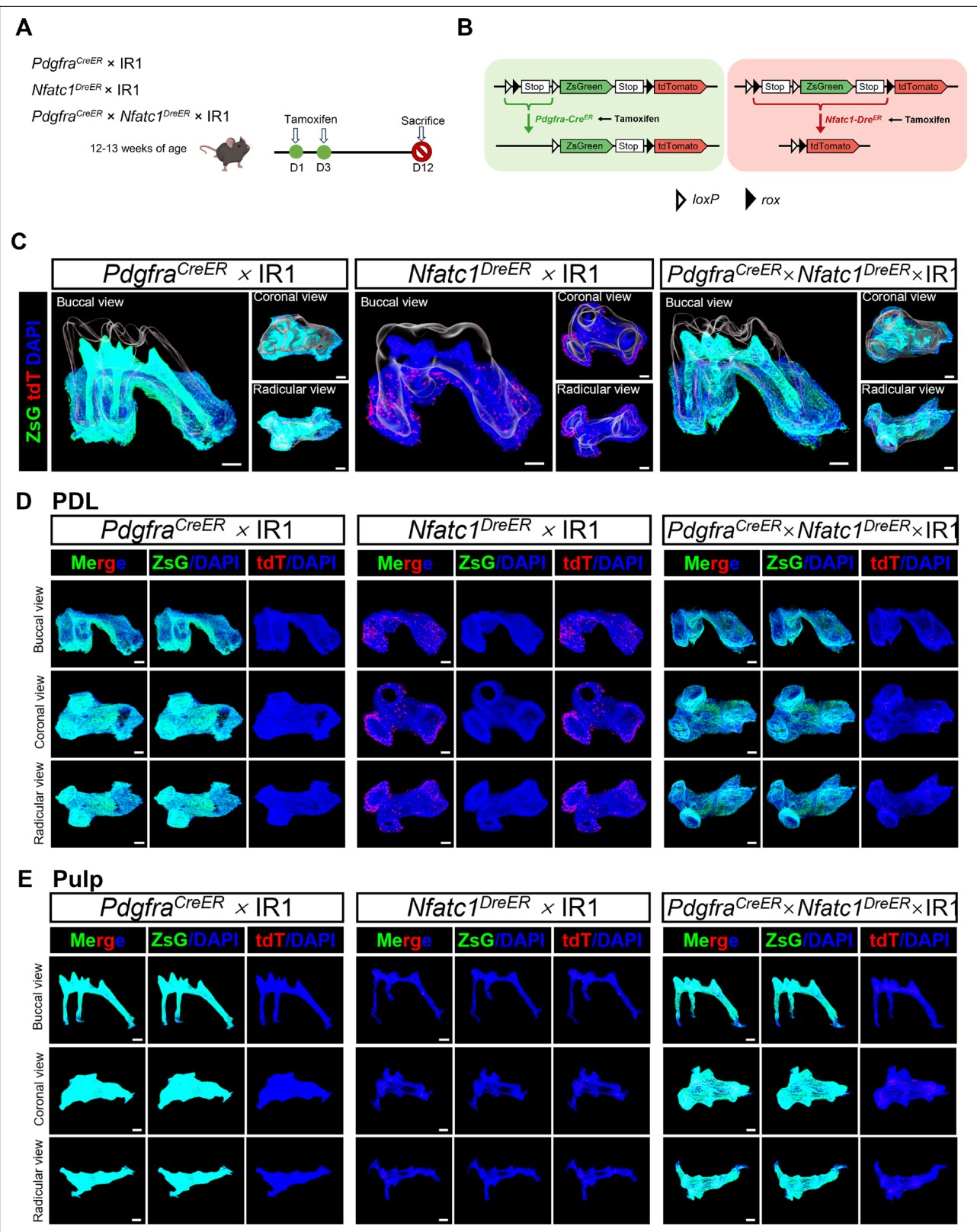

**Figure 8.** IR1 tracing distinct cell populations simultaneously in the pulp and PDL. (**A**) Schematic illustration of lineaging tracing in *Pdgfra^{CreER}*× IR1, *Nfatc1^{DreER}*× IR1 and *Pdgfra^{CreER}*×*Nfatc1^{DreER}*× IR1 mice. (**B**) Schematic diagram of the IR1 working principle. (**C**) 3D reconstruction of maxilla M1 using Imaris, including pulp and PDL with virtual dentin shell (white) (scale bar = 200 μm). The image stack was displayed in buccal view, coronal view, and radicular view of PDL (**D**) and pulp (**E**), respectively.

*Figure 8 continued on next page*

*Figure 8 continued*

The online version of this article includes the following video and figure supplement(s) for figure 8:

**Figure supplement 1.** All consecutive slices (a total of 130 slices) for imaging of maxilla of *Pdgfra^CreER^* × IR1 mice.

**Figure supplement 2.** All consecutive slices (a total of 121 slices) for imaging of maxilla of *Nfatc1^DreER^*× IR1 mice.

**Figure supplement 3.** All consecutive slices (a total of 127 slices) for imaging of maxilla of *Pdgfra^CreER^* ×*Nfatc1^DreER^*× IR1 mice sample.

**Figure supplement 4.** Representative images of coronal pulp (**A**) and PDL (**B**) acquired by confocal microscope of mandible M1 of *Pdgfra^CreER^* × IR1 (**a1, b1**)*, Nfatc1^DreER^*× IR1 (**a2, b2**) and *Pdgfra^CreER^*×*Nfatc1^DreER^*× IR1 (**a3, b3**) mice.

**Figure supplement 5.** 3D reconstruction of maxilla M1 by DICOM-3D of maxilla M1 of *Nfatc1^DreER^*× IR1 and *Pdgfra^CreER^* ×*Nfatc1^DreER^*× IR1 mice.

**Figure 8—video 1.** Panoptic multicolor imaging of ZsGreen⁺ cells in the pulp and PDL area of maxilla M1 of *Pdgfra^CreER^*× IR1 mice, the whole-tissue imaging was reconstructed from serial sections, related to *Figure 8*.

https://elifesciences.org/articles/100173/figures#fig8video1

**Figure 8—video 2.** Panoptic multicolor imaging of tdTomato⁺ cells in the pulp and PDL area of maxilla M1 of *NFATc1^DreER^*× IR1 mice, the whole-tissue imaging was reconstructed from serial sections, related to *Figure 8*.

https://elifesciences.org/articles/100173/figures#fig8video2

**Figure 8—video 3.** Panoptic multicolor imaging of ZsGreen⁺ cells and tdTomato⁺ cells in the pulp and PDL area of maxilla M1 of *Pdgfra^CreER^*×*NFATc1^DreER^*× IR1 mice, the whole-tissue imaging was reconstructed from serial sections, related to *Figure 8*.

https://elifesciences.org/articles/100173/figures#fig8video3

## PDGFR-α⁺ and NFATc1⁺ cells including MSCs and hematopoietic population in pulp and PDL

To identify the population of PDGFR-α⁺ and NFATc1⁺ co-expressing cells in the pulp and periodontal ligament (PDL), we generated *Pdgfra^CreER^*×*Nfatc1^DreER^*× LRTD mice. In these mice, Dre-*rox* and Cre-*loxP* recombination events enable the simultaneous expression of tdTomato and DTR specifically in pulp and PDL cells. In this experiment, we utilized the *R26-CAG-LSL-RSR-tdTomato-2A-DTR* (LRTD) mice solely to label double-positive cells, so DTX was not administered to ablate the cells. To achieve high labeling efficiency, we treated the mice with tamoxifen twice (*Figure 9—figure supplement 1A*). Strong tdTomato signals were detected in both the PDL (*Figure 9—figure supplement 1B*) and pulp (*Figure 9—figure supplement 1C*). With respect to the MSC-specific marker AlphaV, we observed AlphaV⁺tdTomato⁺ cells in both regions. Additionally, CD45⁺ (hematopoietic marker) tdTomato⁺ cells were also present in these areas (*Figure 9—figure supplement 1B,C*). These findings suggest that the population of PDGFR-α⁺ and NFATc1⁺ co-expressing cells is heterogeneous.

## Discussion

PDGFR-α and NFATc1, commonly acknowledged as markers for stem cells, are recognized for their contributions to stem cell characteristics. Our previous study also identified PDGFR-α and NFATc1 cells as SSCs in long bones. The conventional approach to studying the expression patterns of two cell populations involved techniques such as immunofluorescence or immunohistochemical staining, the generation of single-enzyme-cut editing gene mice, or a combination thereof. However, these traditional methods were often susceptible to factors like antibody potency, leading to incomplete and inaccurate depiction of the relationship between the two population of cells. In this study, we have developed two innovative genetic instruments, MCIR and IR1, to facilitate distinct in vivo genetic lineage tracing assays. These tools were specifically designed for precision mapping of cellular lineages and have been utilized, for the first time, within the oral and maxillofacial regions. The arrangement of different recombination sites in relation to reporter genes is crucial for their use in in vivo tracking of multiple cell lineages. The ablation assay demonstrated that PDGFR-α nearly labeled almost mesenchyme cells, which was similar to results of MCIR, noting that MCIR is an available and reliable tool. Our results demonstrate that both IR1 and MCIR are capable of tracing the cell fate mapping of various cells within the oral and craniofacial regions.

Dental pulp and the periodontal supporting tissues surrounding the tooth can be regarded as a relatively integrated biological functional unit. In the research of regeneration and repair mechanisms of dental pulp and PDL, the roles of periodontal and dental pulp stem cells are indispensable (*Ouchi and Nakagawa, 2020*). These cells play a vital role in the oral and maxillofacial regions, not only

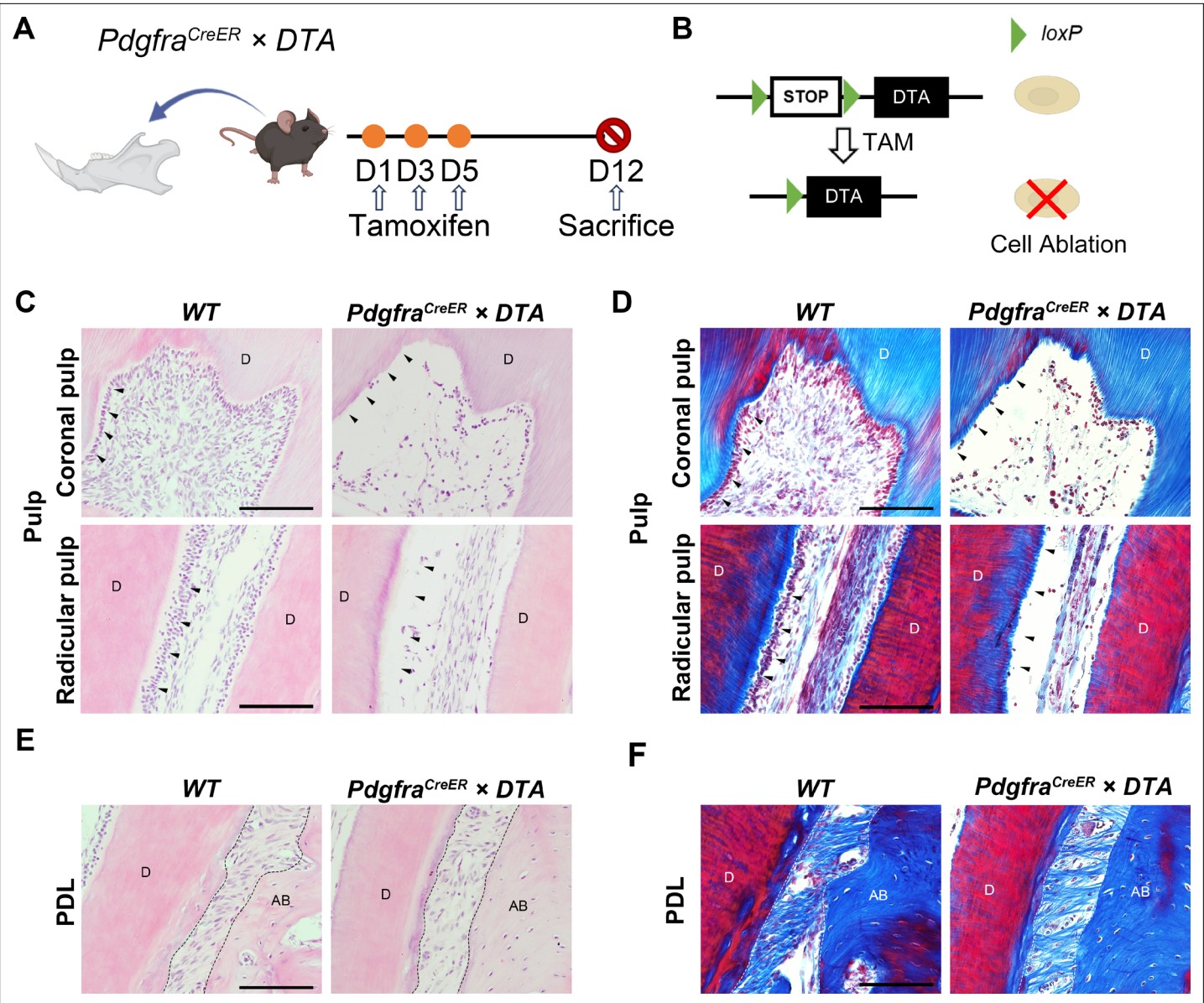

**Figure 9.** Ablation of PDGFR-α⁺ Cells Disrupts the morphology of dental pulp and periodontal ligament tissues. (**A**) Schematics of tamoxifen induction. (**B**) Schematic diagram of the DTA working principle. (**C, E**) Representative H&E images of pulp (C) and PDL (E) of mandible M1 in *Pdgfra^CreER ×DTA* and control mice. (**D, F**) Masson trichrome staining of pulp (D) and PDL (F) of mandible M1 in *Pdgfra^CreER ×DTA* and control mice. Arrows in (**C, D**) indicate the odontoblast cell layer. Dotted lines in (**E, F**) outline ROI of the PDL. **D**: dentin; AB: alveolar bone. Scale bar: 50 μm.

The online version of this article includes the following figure supplement(s) for figure 9:

**Figure supplement 1.** The identification of the types of PDGFR-α⁺ and NFATc1⁺ cells (red) of dental pulp and periodontal ligament tissues.

participating in routine tissue maintenance but also acting as key players in the self-repair process after injury (*Tomokiyo et al., 2019*; *Liang et al., 2020*). Dental pulp stem cells have the potential to regenerate dental pulp tissue, while periodontal stem cells are crucial for the regeneration of periodontal tissues, including alveolar bone, PDL, and gingiva (*Chalisserry et al., 2017*; *Leyendecker Junior et al., 2018*). The interactions between these cell types and their prospects in tissue engineering open new strategies and therapeutic approaches for modern dental treatments.

In our study, MCIR systems have illuminated distinct expression patterns of PDGFR-α⁺, NFATc1⁺ and PDGFR-α⁺ and NFATc1⁺ cells in PDL and dental pulp, indicating their diverse characteristics and roles. In PDL tissue, pulse data demonstrated widespread and abundant expression of PDGFR-α single-positive cells as well as NFATc1 single-positive cells, with no significant alteration in expression

pattern or quantity after lineage tracing. Consequently, we conclude that in periodontal ligament and dental pulp tissues, PDGFR-α single-positive and NFATc1 single-positive cells primarily label intrinsic periodontal mesenchyme in PDL. Conversely, PDGFR-α⁺ and NFATc1⁺ cells exhibited a more confined localization in PDL. The tracing data clearly illustrated that PDGFR-α⁺ and NFATc1⁺ cells successfully gave rise to numerous progenies, which become predominant constituents within the periodontal ligament. In pulp tissue, the distribution of PDGFR-α single-positive cells was similar as that in PDL, primarily labeled odontoblast cell layer and there was not a significant increase in ZsGreen signal after tracing assay (*Figures 5–7*). However, distinct outcomes were observed for NFATc1⁺ single-positive cells, as tracing experiments revealed an increased generation of progeny cells from those homogeneous NFATc1 within the intrinsic dental pulp. Consistently, the results for PDGFR-α⁺ and NFATc1⁺ cells mirrored those of NFATc1⁺ single-positive cells. This observation may imply a niche-specific role for these cells especially PDGFR-α⁺ and NFATc1⁺ cells in maintaining critical functions such as regeneration. This could indicate a more generalized role in tissue maintenance and repair, potentially contributing to the regeneration of both soft and hard tissues.

In order to investigate the hierarchical relationship between PDGFR-α⁺ and NFATc1⁺ in the dental pulp and PDL, we developed IR1 system (*Figure 8*). Our results revealed distinct hierarchical relationships in PDL and dental pulp. In periodontal tissues, compared to the control group (*Pdgfra^CreER*× IR1 and *Nfatc1^DreER*× IR1), there was significantly less tdTomato⁺ cells distribution in the *Pdgfra^CreER*×*Nfatc1^DreER*× IR1 group while the signal of ZsGreen did not decrease significantly. Based on the function of the IR1 system, we could conclude that in the periodontal ligament, PDGFR-α⁺ cells serve as precursor cells for some NFATc1⁺ cells. However, in dental pulp tissue, there was almost no difference in tdTomato⁺ expression between the experimental and control groups, indicating that PDGFR-α⁺ cells cannot be considered as precursor for NFATc1⁺ cells in the dental pulp, or there may not be a clear hierarchical relationship between the two. The sparse presence of NFATc1⁺ cells in both dental pulp and PDL, as observed in the *Nfatc1^DreER*× IR1 group, raised questions about the plasticity and differentiation potential of these cells. Are they primarily quiescent, being activated only upon injury, or do they continuously contribute to tissue homeostasis? Further studies, including functional assays and injury models, would be instrumental in clarifying the dynamic nature of these cells.

Cell ablation and immunofluorescence staining experiments further characterized the types and functions of PDGFR-α⁺/PDGFR-α⁺ and NFATc1⁺ populations. After ablating PDGFR-α⁺ cells, we observed damage to the odontoblast layer and shrinkage of the pulp core in dental pulp tissue, indicating that PDGFR-α⁺ cells contribute to the composition of dental pulp tissue, particularly the odontoblast layer (*Figure 9C and D*). In the periodontal ligament, we noted a reduction and destruction of collagen fibers, suggesting a role for PDGFR-α⁺ cells in periodontal tissue structure (*Figure 9E and F*). Previous results confirmed the presence of double-positive cells in both dental pulp and periodontal tissues and provided insights into their hierarchical relationships in the periodontal ligament (*Figure 8*). To further investigate the double-positive cell population, we developed an inducible dual-editing enzyme reporter system to label these cells with tdTomato signals. Using AlphaV as a marker for mesenchymal stem cells (MSCs) and CD45 for hematopoietic cells, we found that double-positive cells included components of both MSCs and hematopoietic cells (*Figure 9—figure supplement 1B,C*), indicating a heterogeneous population. Further experiments are necessary to determine whether the predominant role in this co-positive MSC population is played by PDGFR-α⁺ or NFATc1⁺ and to clarify the specific functions of these cells in the future.

For the traditional sectioning & confocal imaging procedure, it can obtain a highly distinct single-layer image. However, it is troubled to determine whether the different cell populations overlap in spatial location. At the same time, the experimental results were deeply affected by the differences between different section, which may cause error. It is also impossible to determine the amount and spatial position of a certain cell population in the whole tissue. In this study, we utilized two different techniques to map the distribution of aimed cells. We successfully achieved a 3D cellular-level visualization of PDL and pulp from *Pdgfra^CreER*×*Nfatc1^DreER*× LGRT mice by TC-based advanced imaging and 3D reconstructions based on continuous sections provide a more accurate representation of spatial distribution.TC-based advanced imaging procedure can clearly visualize its 3D structure, reconstruct the whole across latitudes, and understand the spatial position and expression of each structure, which could avoid the bias of traditional single-layer slicing may cause, and provides a more intuitive and objective description of the existing situation. However, our results demonstrated TC still has

some limitations. During the experiment, we found that the signal of tdTomato in the pulp and PDL area was poor and could not be distinguished from autofluorescence through the imaging of light-sheet system, which may due to the limitations of editing efficiency in DNA recombinase–mediated lineage-tracing system, which guaranteed weaker tdTomato signal compared to Zsgreen. In addition, the total amount of NFATc1$^+$ SSCs in these regions was limited, with the highly distribution in the alveolar bone, jaw bones, and trabecular bone surface edge, which guaranteed less visible signal in the target region. Another possibility is that the as-describe TC method may cause poor penetration of the tdTomato fluorescence signals. The above results prompted that we should minimize the sample processing time to avoid fluorescence depletion to the greatest extent when using dual DNA recombinases-based genetic lineage tracing system for whole tissue imaging after TC treatment.

The 3D sections reconstruction results, however, effectively addressed the issue of weak tdTomato signal and provide a clearer visualization of the distribution of ZsGreen and tdTomato signals. For example, the tdTomato signal in the root pump, which was almost completely unobservable by TC-based imaging, can be clearly seen using confocal imaging and 3D reconstruction (*Figures 3C–D and 6C–D*, and *Figure 3—figure supplement 3*, *Figure 6—figure supplement 3*). However, compared to TC, the quality of 3D reconstruction of sections still relies on the angle and quality of the sections, with the section angle having a significant impact on the reconstruction outcome. In addition, because the slice itself has a certain thickness (10 µM in this study), which leads to the appearance of discontinuous in the final reconstructed image, and the aesthetics and accuracy could be affected to a certain extent. Also, unavoidable tissue damage during the sectioning process may result in the loss of some information. Therefore, a variety of different information could be obtained through two different imaging technologies, which prompt us to use the advanced experimental procedure according to the actual purpose.

The use of advanced genetic tools has also shed light on the limitations and challenges associated with current lineage tracing methodologies (*Liu et al., 2013*). For instance, the potential for recombinase leakiness or off-target effects necessitates the inclusion of stringent controls and corroborative experimental approaches (*Zhao and Zhou, 2019*). As such, the refinement of these tools and the development of additional specific markers are required to enhance the accuracy of cell fate mapping.

In conclusion, our study developed multiple DNA recombinases based on genetic lineage tracing and improved their accuracy and usability cooperating with advanced imaging techniques in order to map the atlas of NFATc1$^+$/ PDGFR-α$^+$ inclusive, exclusive and hierarchical distribution in dental and periodontal mesenchyme. Our study underscored the importance of NFATc1$^+$ and PDGFRα$^+$ cells in the homeostasis and regeneration of oral and maxillofacial tissues. The nuanced understanding of their distribution and potential roles paves the way for novel regenerative strategies that could harness the intrinsic capabilities of these cells. Future research should focus on the functional characterization of these cells, their responsiveness to injury, and the interplay of signaling pathways that regulate their activity, which will be crucial for translating these findings into clinical applications.

## Materials and methods

### Ethics statement and animals

All animal procedures were reviewed and approved by Ethical Committees of West China School of Stomatology, Sichuan University (WCHSIRB-D-2017–041). Relative surgical models were conducted according to approved guidelines set by State Key Laboratory of Oral Diseases, West China Hospital of Stomatology.

*Nfatc1$^{DreER}$* and IR1 strain were kindly provided by Prof. Bin Zhou (Chinese Academy of Sciences), which were also reported in our previous study (*Yu et al., 2022*). *Pdgfra$^{CreER}$* (Stock 018280) was purchased from the JAX Lab, *H11-CAG-LSL-ZsGreen-CAG-RSR-tdTomato* and *R26-CAG-LSL-RSR-tdTomato-2A-DTR* was purchased from Shanghai Model Organisms Center (Strain# NM-KI-200319 and NM-KI-190086 respectively), and DTA mice were purchased from JAX Lab (Stock# 009669).

### In vivo mouse studies

All the transgenic mice discussed in this work are adults around 12–14 weeks, which were intraperitoneally injected with tamoxifen (TAM) according to our previous report (*Yu et al., 2022*).

## RT-qPCR

For RT-qPCR total RNA was extracted using the TRIzolTM (Invitrogen) according to the manufacturer's protocol. Complementary DNA was synthesized by using the HiScript III RT SuperMix for qPCR (Vazyme) in accordance to user manuals. Then quantitative real-time PCR was performed in triplicate by using AceQ Universal SYBR qPCR Master Mix (Vazyme) for PCR reactions on an iCycler Real-Time Detection System (BioRad, USA).

## Maxillae and cranium collection, fixation, and decalcification for light sheet imaging

Fresh dissected tissues were washed with PBS (Solarbio), and then rapidly transferred to 4% PFA (Sigma-Aldrich) for 4 hr. Before fixation, the muscle and fat attached to the tissue surface should be removed as much as possible, otherwise there will be strong interference signal during imaging. The tissues were then washed three times with PBS at 4 °C on a rocker platform for 15 min each time. For decalcification, tissues were incubated with 0.5 M EDTA solution (pH 7.4) at 4 °C for 24 hr. After decalcification, the samples were washed three times with PBS on the rocker (15 mi each time). The tissues were then submersed in 50%, 80% and 100% ethanol gradient for 30 min each to dehydrated. 100% ethanol was changed twice after every 20 min. Then the samples were immersed in 5% (V/V) H2O2 for 2 hr (Sigma-Aldrich, H1009) for bleaching (*Biswas et al., 2023*).

## Clearing of maxillae and cranium

After bleaching, the samples were rinsed three times with Ethyl cinnamate (ECi) (Sigma-Aldrich, 112372) for 5 min each at room temperature.

## Histological preparation and staining

Specifically, the tissues were fixed in 4% PFA at 4°C for 24 hr. After fixation, the maxilla was immersed in 30% sucrose solution at 4 °C overnight. Then the samples were embedded in OCT tissue freezing medium (Leica) and placed on liquid nitrogen to conduct non-decalcified frozen sectioning at 7–10 μm thickness. The slices were blocked with 5% BSA in PBST for 20 min. The microspheres were then incubated with the appropriate primary antibodies (Mouse monoclonal anti-AlphaV, Santa Cruz, Cat#sc-376156; CD45 Rabbit mAb, Cell Signaling Technology, Cat#70257) overnight at 4 °C. On the following day, the microspheres were washed three times with fresh PBS and incubated with the specific secondary antibody, with or without DAPI, for 2 hr at room temperature. Finally, the microspheres underwent three washes with PBS, each lasting 10 min, were mounted with anti-fade fluorescence mounting medium, and were then sealed for detection. For H&E staining and Masson trichrome staining, decalcified samples were dehydrated in graded ethanol and embedded in paraffin. Mandibles were sectioned into 6 μm slices using the microtome (Leica RM2255). H&E staining was performed according to the manufacturer's instruction (Biosharp). Masson trichrome staining was performed according to the manufacturer's instruction (Solarbio).

## Imaging and 3D reconstruction of maxillae and mandible sections

Digital pathological system (Olympus FV3000) was used to scan all the stained sections and reconstructed by following the previously described protocol (Wu et al. 2020). The light sheet fluorescence microscopy-Zeiss Lightsheet Z.7 was used to acquire clearing samples microscopic image stacks.

## Image analysis and quantifications

Slices/Z-stacks of images acquired on the light sheet and confocal microscope were processed and reconstructed in three dimensions with Imaris software (version 10.0.1). Imaris, Adobe Photoshop 25.0, and ImageJ were used for image processing and analysis in line with the journal's guidance for image processing. Quantification of cell numbers was done on a z stack of images in Imaris using the automatic spot detection feature.

## 3D surface reconstruction

3D surface rendering in images was applied using the surface module in Imaris.

## Statistical analysis

All data were presented as the mean ± SEM. A two-tailed Student's t-test was used for comparison between groups. $p < 0.05$ was considered statistically significant. N.S. stands for not significant.

## Acknowledgements

This work was supported by National Key Research and Development Program of China 2023YFC3605600 (LY and FY); National Natural Science Foundation of China 82201045 (FY) and 82100982 (FL); and Young Elite Scientist Sponsorship Program by CAST (2022QNRC001 to FY).

## Additional information

### Funding

| Funder | Grant reference number | Author |
| --- | --- | --- |
| National Key Research and Development Program of China | 2023YFC3605600 | Ling Ye Fanyuan Yu |
| National Natural Science Foundation of China | 82201045 | Fanyuan Yu |
| National Natural Science Foundation of China | 82100982 | Feifei Li |
| Young Elite Scientist Sponsorship Program by CAST | 2022QNRC001 | Fanyuan Yu |

The funders had no role in study design, data collection and interpretation, or the decision to submit the work for publication.

### Author contributions

Xue Yang, Data curation, Software, Investigation, Visualization, Methodology, Writing – original draft; Chuyi Han, Data curation, Software, Validation, Investigation, Methodology, Writing – original draft, Writing – review and editing; Changhao Yu, Software, Methodology; Bin Zhou, Supervision, Validation, Methodology; Ling Ye, Resources, Supervision, Funding acquisition; Feifei Li, Conceptualization, Resources, Supervision, Funding acquisition; Fanyuan Yu, Conceptualization, Resources, Supervision, Funding acquisition, Validation, Methodology, Project administration, Writing – review and editing

### Author ORCIDs

Fanyuan Yu https://orcid.org/0000-0002-6879-3508

### Ethics

All animal procedures were reviewed and approved by Ethical Committees of West China School of Stomatology, Sichuan University (WCHSIRB-D-2017-041). Relative surgical models were conducted according to approved guidelines set by State Key Laboratory of Oral Diseases, West China Hospital of Stomatology.

Reviewer #1 (Public review): https://doi.org/10.7554/eLife.100173.3.sa1
Author response https://doi.org/10.7554/eLife.100173.3.sa2

## Additional files

### Supplementary files

- MDAR checklist
- Supplementary file 1.

## Data availability

All data generated or analysed during this study are included in the manuscript and supporting files; source data file has been provided for Figure 5-figure supplement 1.

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
