## [Editor Report · eLife Assessment]

Utilizing transgenic lineage tracing techniques and tissue clearing-based advanced imaging and three-dimensional slices reconstruction, the authors comprehensively mapped the distribution atlas of NFATc1+ and PDGFR-α+ cells in dental and periodontal mesenchyme and tracked their in vivo fate trajectories. This **important** work extends our understanding of NFATc1+ and PDGFR-α+ cells in dental and periodontal mesenchyme homeostasis, and should provide impact on clinical application and investigation. The strength of this work is **compelling** in employing CRISPR/Cas9-mediated gene editing to generate two dual recombination systems, and mapped gNFATc1+ and PDGFR-α+ cells residing in dental and periodontal mesenchyme, their capacity for progeny cell generation, and their inclusive, exclusive and hierarchical relations in homeostasis, generating a spatiotemporal atlas of these skeletal stem cell population.

---

## [Referee Report · Reviewer #1 (Public review)]

Summary:

Utilizing transgenic lineage tracing techniques and tissue clearing-based advanced imaging and three-dimensional slices reconstruction, the authors comprehensively mapped the distribution atlas of NFATc1+ and PDGFR-α+ cells in dental and periodontal mesenchyme and tracked their in vivo trajectories. This important work expands our understanding of both single and double positive NFATc1 and PDGFR-α cells in maintaining dental and periodontal mesenchyme homeostasis, and will provide impact on clinical application and investigation. The strength of this work is convincing, as it employed CRISPR/Cas9-mediated gene editing to generate two dual recombination systems, and mapped gNFATc1+ and PDGFR-α+ cells residing in dental and periodontal mesenchyme, their capacity for progeny cell generation, and their inclusive, exclusive and hierarchical relations in homeostasis, generating a spatiotemporal atlas of these skeletal stem cell population.

This work has theoretical or practical implications in the periodontal field. The methods, data and analyses support the claims.

Comments on revised version:

The authors have addressed my main concerns.

---

## [Author Response]

The following is the authors’ response to the original reviews.

**Public Reviews:**

**Reviewer #1 (Public Review):**
In this study, Yang et al. investigated the locations and hierarchies of NFATc1+ and PDGFRα+ cells in dental and periodontal mesenchyme. By combining intersectional and exclusive reporters, they attempted to distinguish among NFATc1+PDGFRα+, NFATc1+PDGFRα-, and NFATc1- PDGFRα+ cells. Using tissue clearing and serial section-based 3D reconstruction, they mapped the distribution atlas of these cell populations. Through DTA-induced ablation of PDGFRα+ cells, they demonstrated the crucial role of PDGFRα+ cells in the formation of the odontoblast cell layer and periodontal components.

Thank you for your valuable comments and suggestions, which have greatly enhanced the quality of this research article. The manuscript has been significantly revised in accordance with the reviewers’ comments. All necessary experimental conditions and required data have been included, and all the questions and considerations have been well-addressed in the revised manuscript and supporting information.

Main issues:(1) The authors did not quantify the contribution of PDGFRα+ cells or NFATc1+ cells to dental and periodontal lineages in PDGFRαCreER; Nfatc1DreER; LGRT mice. Zsgreen+ cells represented PDGFRα+ cells and their lineages. Tomato+ cells represented NFATc1+ cells and their lineages. Tomato+Zsgreen+ cells represented NFATc1+PDGFRα+ cells and their lineages. Conducting immunostaining experiments with lineage markers is essential to determine the physiological contributions of these cells to dental and periodontal homeostasis.

Thanks for your question, we are sorry for the insufficient statement. Figure S9 provided statistical analysis of the number of PDGFR-α+ cells, NFATc1+ cells, and PDGFR-α+&NFATc1+ cells in the dental pulp and periodontal ligament (PDL). The results allow for a clear comparison of the contributions of single-positive and double-positive cells to both tissues. Additionally, the tracing results showed whether these three cell populations have the capacity to produce progeny cells. We further supplemented the analysis with immunofluorescence results of double-positive cells to identify their cell types, selecting AlphaV as the marker for mesenchymal stem cells (MSCs) and CD45 as the marker for hematopoietic cells. This part is further discussed in the manuscript as below:

Page 14-15 in the revised manuscript, “To identify the population of PDGFR-α+ and NFATc1+ co-expressing cells in the pulp and periodontal ligament (PDL), we generated *PdgfraCreER*; *Nfatc1DreER*; R26-LSL-RSR-tdT-DTR (LRTD) mice... Strong tdTomato signals were detected in both the PDL (Figure S22B) and pulp (Figure S22C). With respect to the MSC-specific marker AlphaV, we observed AlphaV+tdTomato+ cells in both regions. Additionally, CD45+ (hematopoietic marker) tdTomato+ cells were also present in these areas (Figure S22B, C). These findings suggest that the population of PDGFR-α+ and NFATc1+ co-expressing cells is heterogeneous.”

(2) The authors attempted to use PDGFRαCreER; Nfatc1DreER;IR1 mice to illustrate the hierarchies of NFATc1+ and PDGFRα+ cells. According to the principle of the IR1 reporter, it requires sequential induction of PDGFRα-CreER and Nfatc1-DreER to investigate their genetic relationship. Upon induction by tamoxifen, NFATc1+PDGFRα- cells and NFATc1-PDGFRα+ cells were labeled by Tomato and Zsgreen, respectively. However, the reporter expression of NFATc1+PDGFRα+ cells was uncertain, most likely random. Therefore, the hierarchical relationship of NFATc1+ and PDGFRα+ cells cannot be reliably determined from PDGFRαCreER; Nfatc1DreER; IR1 mice.

Thank you for your question. We have supplemented the control group (*PdgfraCreER; IR1)* experimental data (Figure 8). By comparing the results of *PdgfraCreER; Nfatc1DreER; LGRT* tracing assays, we confirmed that the expression pattern and range of PDGFR-a+ cells in pulp and PDL of *PdgfraCreER; IR1* mice are consistent with those observed in *PdgfraCreER × Nfatc1DreER ×* LGRT mice (Figure 6), and the same applies to NFATc1+ cells. All of our experimental results have been repeated multiple times. In addition, the IR1 system was initially developed by Professor Bin Zhou's lab and was validated for feasibility and stability in a paper published in *Nature Medicine* in 2017 (https://doi.org/10.1038/nm.4437). Moreover, Professor Zhou Bo O's team applied IR1 dual recombinases for bone lineage tracing in 2021 published in *Cell Stem Cell*, which also confirmed its feasibility and stability. (DOI: 10.1016/j.stem.2021.08.010)

**Reviewer #2 (Public Review):**
Summary:Yang et al. present an article investigating the spatiotemporal atlas of NFATc1+ and PDGFR-α+ cells within the dental and periodontal mesenchyme. The study explores their capacity for progeny cell generation and their relationships - both inclusive and hierarchical - under homeostatic conditions. Utilizing the Cre/loxP-Dre/Rox system to construct tool mice, combined with tissue transparency and continuous tissue slicing for 3D reconstruction, the researchers effectively mapped the distribution of NFATc1+ and PDGFR-α+ cells. Additionally, in conjunction with DTA mice, the study provides preliminary validation of the impact of PDGFR-α+ cells on dental pulp and periodontal tissues. Primarily, this study offers an in-situ distribution atlas for NFATc1+ and PDGFR-α+ cells but provides limited information regarding their origin, fate differentiation, and functionality.

We would like to thank the reviewer for setting a high value on our study. Given many constructive suggestions, the manuscript has been revised to improve the quantity of this study. All the necessary discussions have also been added, and all the questions and concerns have been well-addressed in the revised manuscript. The point-to-point reply to the comments is listed below:

Strengths:(1) Tissue transparency techniques and continuous tissue slicing for 3D reconstruction, combined with transgenic mice, provide high-quality images and rich, reliable data.(2) The Cre/loxP and Dre/Rox systems used by the researchers are powerful and innovative.(3) The IR1 lineage tracing model is significantly important for investigating cellular differentiation pathways.(4) This study provides effective spatial distribution information of NFATc1+/PDGFR-α+ cell populations in the dental and periodontal tissues of adult mice.Weaknesses:(1) In the functional experiment section, the investigation into the role of NFATc1+/PDGFR-α+ cell populations is somewhat lacking.

Thank you so much for your comments and suggestions. We have supplemented the analysis with immunofluorescence results of double-positive cells to identify NFATc1+&PDGFR-α+ cell populations, selecting AlphaV as the marker for mesenchymal stem cells (MSCs) and CD45 as the marker for hematopoietic cells. This part was shown as below:

Page 14-15 in the revised manuscript, “To identify the population of PDGFR-α+ and NFATc1+ co-expressing cells in the pulp and periodontal ligament (PDL), we generated *PdgfraCreER*; *Nfatc1DreER*; *R26-LSL-RSR-tdT-DTR* (LRTD) mice… Strong tdTomato signals were detected in both the PDL (Figure S22B) and pulp (Figure S22C). With respect to the MSC-specific marker AlphaV, we observed AlphaV+tdTomato+ cells in both regions. Additionally, CD45+ (hematopoietic marker) tdTomato+ cells were also present in these areas (Figure S22B, C). These findings suggested that the population of PDGFR-a+ and NFATc1+ co-expressing cells is heterogeneous.”

We also supplemented the discussion regarding the role of PDGFR-α+ population on page 17. Its potential role in pulp and periodontal formation had been suggested as well.

Page 17 in the revised manuscript, “After ablating PDGFR-α+ cells, we observed damage to the odontoblast layer and shrinkage of the pulp core in dental pulp tissue, indicating that PDGFR-α+ cells contribute to the composition of dental pulp tissue, particularly the odontoblast layer (Figure. 9C, D). In the periodontal ligament, we noted a reduction and destruction of collagen fibers, suggesting a role for PDGFR-α+ cells in periodontal tissue structure (Figure. 9E, F).”

(2) The author mentions that 3D reconstruction of consecutive tissue slices can provide more detailed information on cell distribution, so what is the significance of using tissue-clearing techniques in this article?

Thank you for your insightful comment, and we are sorry for the insufficient statement here. In our study, the utilization of tissue clearing techniques was to address some of the shortcomings associated with the 3D reconstruction of consecutive tissue slices, such as the compromised integrity of samples due to section layering, leading to discontinuities along the z-axis and potential loss of positive signals (Fig. S5, S13). Additionally, unavoidable tissue damage during the sectioning process may result in the loss of some information. As one of the most advanced imaging technologies currently available, tissue clearing/imaging allows for direct observation of the spatial location and relationships of fluorescently labeled cells within the intact tissue, which is more persuasive. Also, evolving beyond the analysis of structural and molecular biology of selected tissue sections, and expanding the focus to entire organs and organisms, is a trend in the development of the biomedical field (Nat Methods. 2024 Jul;21(7):1153-1165; Nat Commun. 2024 Feb 26;15(1):1764). Admittedly, no method is flawless; thus, our employment of two advanced imaging approaches aims to answer questions regarding the spatial positioning and relationships of PDGFR-α single-positive, NFATc1 single-positive cells, and PDGFR-α+ NFATc1+ cells from multiple perspectives. This is done to enhance the credibility and persuasiveness of our results.

We greatly appreciate your suggestion, which have significantly complemented the content of our article. The corresponding statements have been added in the revised manuscript as below:

Page 6 in the revised manuscript, “As one of the most advanced imaging technologies currently available, tissue clearing/imaging allows for direct observation of the spatial location and relationships of fluorescently labeled cells within the intact tissue. Therefore, according to the existing SUMIC tissue deep clearing (TC) methods, we modified and improved a rapid and efficient procedure, which enable rapid single-cell resolution and quantitative panoptic 3D light-sheet imaging.”

(3) After reading the entire article, it is confusing whether the purpose of the article is to explore the distribution and function of NFATc1+/PDGFR-α+ cells in teeth and periodontal tissues, or to compare the differences between tissue clearing techniques and 3D reconstruction of continuous histological slices using NFATc1+/PDGFR-α+ cells?

We sincerely appreciate your question and apologize for any ambiguous descriptions.

The purpose of our study is to map the atlas of NFATc1+/ PDGFR-α+ inclusive, exclusive and hierarchical distribution in dental and periodontal mesenchyme. Under this premise, the two advanced imaging techniques were merely employed as means to elucidate this issue Indeed, in the previous manuscript, we did overemphasize the comparison and description of the differences between tissue clearing techniques and 3D reconstruction of continuous slices, which led to unnecessary misunderstandings for which we are deeply apologetic. Consequently, in this version of the manuscript, we have diminished the descriptions comparing their advantages and disadvantages, focusing instead on exploring the importance of NFATc1+/PDGFR-α+ cells. We appreciate your suggestions once again.

Page 6 in the revised manuscript, “These two 3D-reconstruction and imaging technologies complement each other to jointly address the spatial positioning and hierarchical relationships of PDGFR-α+, NFATc1+, and PDGFR-α+ NFATc1+ cells from multiple perspectives.”

(4) The researchers did not provide a clear definition of the cell types of NFATc1+/PDGFR-α+ cells in teeth and periodontal tissues.

Thanks for your suggestions. We discovered through cell ablation experiments that the removal of PDGFR-α+ cells resulted in the destruction of the odontoblast layer in the dental pulp, shrinkage of the pulp core, and disruption of collagen fibers in the periodontal ligament. Combined with the results from lineage tracing, we conclude that PDGFR-α+ cells primarily constitute the mesenchymal cells that form the supporting tissues in both the dental pulp and periodontal ligament (Part 4.1). Through immunofluorescence staining, AlphaV was as the marker for mesenchymal stem cells (MSCs) and CD45 as the marker for hematopoietic cells, we observed that the double-positive cell population was a heterogeneous group, containing both mesenchymal stem cells (MSC) and hematopoietic cells (Part 4.2).

(5) In studies related to long bones, the author defines the NFATc1+/PDGFR-α+ cell population as SSCs, which as a stem cell group should play an important role in tooth development or injury repair. However, the distribution patterns and functions of the NFATc1+/PDGFR-α+ cell population in these two conditions have not been discussed in this study.

Thanks for your suggestions. The NFATc1+/PDGFR-α+ cell population was identified as playing an important role in tissue regeneration, especially in oral and maxillofacial tissues. Our research primarily focuses on the identification of NFATc1+ and PDGFR-α+ cells within dental and periodontal mesenchyme, highlighting their contribution to tissue homeostasis and regeneration. Although the NFATc1+/PDGFR-α+ cells were characterized in the context of other tissue types, their detailed role in tooth development and injury repair remains an area for further exploration.

This part was further discussed on page 17-18 in the revised manuscript, “Cell ablation and immunofluorescence staining experiments further characterized the types and functions of PDGFR-α+/PDGFR-α+&NFATc1+ populations. After ablating PDGFR-α+ cells, we observed damage to the odontoblast layer and shrinkage of the pulp core in dental pulp tissue, indicating that PDGFR-α+ cells contribute to the composition of dental pulp tissue, particularly the odontoblast layer (Figure. 9C, D). In the periodontal ligament, we noted a reduction and destruction of collagen fibers, suggesting a role for PDGFR-α+ cells in periodontal tissue structure (Figure. 9E, F). Previous results confirmed the presence of double-positive cells in both dental pulp and periodontal tissues and provided insights into their hierarchical relationships in the periodontal ligament (Figure. 8). To further investigate the double-positive cell population, we developed an inducible dual-editing enzyme reporter system to label these cells with tdTomato signals. Using AlphaV as a marker for mesenchymal stem cells (MSCs) and CD45 for hematopoietic cells, we found that double-positive cells included components of both MSCs and hematopoietic cells (Figure S22B, C), indicating a heterogeneous population. Further experiments are necessary to determine whether the predominant role in this co-positive MSC population is played by PDGFR-α+ or NFATc1+ and to clarify the specific functions of these cells in the future.”

**Reviewer #3 (Public Review):**
Summary:This groundbreaking study provided the most advanced transgenic lineage tracing and advanced imaging techniques in deciphering dental/periodontal mesenchyme cells. In this study, authors utilized CRISPR/Cas9-mediated transgenic lineage tracing techniques to concurrently demonstrate the inclusive, exclusive, and hierarchical distributions of NFATc1+ and PDGFR-α+ cells and their lineage commitment in dental and periodontal mesenchyme.Strengths:In cooperating with tissue clearing-based advanced imaging and three-dimensional slices reconstruction, the distribution and hierarchical relationship of NFATc1+ and PDGFR-α+ cells and progeny cells plainly emerged, which undoubtedly broadens our understanding of their in vivo fate trajectories in craniomaxillofacial tissue. Also, the experiment design is comprehensive and well-executed, and the results are convincing and compelling.Weaknesses:Minor modifications could be made to the paper, including more details on the advantages of the methodology used by the authors in this study, compared to other studies.

Thanks for your constructive comments and advice on how to improve the quality of this research article. We have thoroughly and carefully corrected the manuscript based on your suggestion, and all the necessary data have been added to support our claims. Meanwhile, all the questions and concerns have been well-addressed in the revised manuscript and the revised supplementary information. Thus, we believe that the quality of this paper has been significantly enhanced. We thank you again for your great efforts.

**Recommendations For The Authors:**

**Reviewer #1 (Recommendations For The Authors):**
(1) Line 134, the authors categorized the reporter systems into three types: intersectional reporters, exclusive reporters, and nested reporters. However, Figure 1A does not depict the nested reporters.

Thanks for your helpful recommendation to improve the quality of this manuscript, and we are sorry for the mistake. In this revised manuscript, we have modified the content of Figure 1A, as displayed below:

(2) Line 238, the authors mentioned that NFATc1 is expressed in the mandible and periodontal tissues based on their previous sequencing analyses. It would be better to cite the related reference or display the expression of NFATc1 in the Supplemental Figures.

Thanks for your suggestions. We sincerely apologize for the typo that occurred during the writing process and have revised the original text to on page 9:

“The previous sequencing analyses have reported the expression of NFATc1 in mandible and periodontal tissues20. (DOI: 10.1177/00220345221074356)”

(3) Line 264, the figure callout "Figure 5E" does not exist, and the figure legends of Figure 5 contain the same error.

We greatly appreciate your rigor and diligence, and we have corrected this error.

(4) Line 280, the figure callout "Figure S12" is incorrect.

Thank you for your efforts, and we are sorry for our negligence. The corresponding descriptions have been amended as below:

Page 10 in the revised manuscript, “Consistent with the quantification of TC-based imaging results (Figure S9), the number of PDGFR-α+ cells and NFATc1+ cells were significantly higher than that in pulse group.”

(5) Line 301, the figure callout "Figure 4" is erroneous.

Thank you for your efforts, and we are sorry for our negligence. The corresponding descriptions have been amended as below:

Page 11 in the revised manuscript, “After 11 days tracing, the number of PDGFR-α+ & NFATc1+ cells and PDGFR-α+NFATc1+ cells increased significantly (Figure 7)…”

(6) Line 306, the sentence "Our previous study identified the presence of NFATc1+ cells in the cranium by single-cell sequencing (unpublished data)" could be improved by referencing specific data or findings.

Thanks for your suggestions, and we are sorry for our negligence. The corresponding citation have been amended as below:

Page 11 in the revised manuscript, “As a part of craniomaxillofacial hard tissue, we also intended to explore whether the presence of NFATc1+ and PDGFR-α+ cells in cranial bone tissue/suture is different from dental and periodontal tissue (our previous study has identified the presence of NFATc1+ cells in the cranium by single-cell sequencing28”)

(7) Line 341, the statement "Moreover, no PDGFR-α+ cells were detected in the Nfatc1DreER; IR1 group," needs further explanation or context.

Thanks for your suggestions. The corresponding descriptions have been amended as below:

Page 13 in the revised manuscript, “Moreover, since the recombinase recognition sites are interleaved (*loxP–rox–loxP–rox*), recombination by one system will naturally remove a recognition site of the other system, rendering its reporter gene inactive for further recombination. The results showed no tdTomato+ cells or ZsGreen+ cells were detected in the *PdgfraCreER ×* IR1 or *Nfatc1DreER ×* IR1 group respectively demonstrating the feasibility and accuracy of the IR1 system.”

(8) Several statements in this text were duplicated. For instance, lines 365 to 376 are identical to lines 497 to 508. This redundancy should be addressed to improve the manuscript's clarity and conciseness.

We greatly appreciate your suggestions, and we are sorry for the misunderstanding we may have caused. We have revised and integrated the entire Results 4 section (including lines 365 to 376 of the original manuscript) into the Discussion section to avoid unnecessary redundancy and misunderstandings. This adjustment also emphasizes that the goal of using two imaging techniques is to draw more credible conclusions from multiple perspectives, thereby mitigating the shortcomings of relying solely on existing advanced imaging methods. The revised content are as follows:

Page 18 in the revised manuscript, “TC-based advanced imaging procedure can clearly visualize its 3D structure, reconstruct the whole across latitudes, and understand the spatial position and expression of each structure, which could avoid the bias of traditional single-layer slicing may cause, and provides a more intuitive and objective description of the existing situation. However, our results demonstrated TC still has some limitations…”

Page 19 in the revised manuscript, “The 3D sections reconstruction results, however, effectively addressed the issue of weak tdTomato signal and provide a clearer visualization of the distribution of ZsGreen and tdTomato signals. For example, the tdTomato signal in the root pump, which was almost completely unobservable by TC-based imaging, can be clearly seen using confocal imaging and 3D reconstruction (Figure 3C-D, Figure 6C-D, and Figure S4, Figure S12). However, compared to TC, the quality of 3D reconstruction of sections still relies on the angle and quality of the sections, with the section angle having a significant impact on the reconstruction outcome. In addition, because the slice itself has a certain thickness (10 μM in this study), which leads to the appearance of discontinuous in the final reconstructed image, and the aesthetics and accuracy could be affected to a certain extent. Also, unavoidable tissue damage during the sectioning process may result in the loss of some information. Therefore, a variety of different information could be obtained through two different imaging technologies, which prompt us to use the advanced experimental procedure according to the actual purpose.”

**Reviewer #2 (Recommendations For The Authors):**
(1) It should be further highlighted in the article what cell type the NFATc1+/PDGFR-α+ cells should be defined as in teeth and periodontal tissues.

Thank you so much for your suggestions. We have supplemented the analysis with immunofluorescence results of double-positive cells to identify NFATc1+&PDGFR-α+ cell populations, selecting AlphaV as the marker for mesenchymal stem cells (MSCs) and CD45 as the marker for hematopoietic cells.

This part was on page 14-15 in the revised manuscript, “To identify the population of PDGFR-α+ and NFATc1+ co-expressing cells in the pulp and periodontal ligament (PDL), we generated *PdgfraCreER*; *Nfatc1DreER*; *R26-LSL-RSR-tdT-DTR (*LRTD*)* mice… Strong tdTomato signals were detected in both the PDL (Figure S22B) and pulp (Figure S22C). With respect to the MSC-specific marker AlphaV, we observed AlphaV+tdTomato+ cells in both regions. Additionally, CD45+ (hematopoietic marker) tdTomato+ cells were also present in these areas (Figure S22B, C). These findings suggested that the population of PDGFR-a+ and NFATc1+ co-expressing cells is heterogeneous.”

We also supplemented the discussion regarding the role of PDGFR-α+ population on page 17. Its potential role in pulp and periodontal formation had been suggested as well:

Page 17 in the revised manuscript: “After ablating PDGFR-α+ cells, we observed damage to the odontoblast layer and shrinkage of the pulp core in dental pulp tissue, indicating that PDGFR-α+ cells contribute to the composition of dental pulp tissue, particularly the odontoblast layer (Figure. 9C, D). In the periodontal ligament, we noted a reduction and destruction of collagen fibers, suggesting a role for PDGFR-α+ cells in periodontal tissue structure (Figure. 9E, F).”

(2) The authors are advised to supplement the description of the cellular origin and the differentiation trajectory of NFATc1+/PDGFR-α+ cells in teeth and periodontal tissues.

Thank you for your suggestion. Our study currently focused more on mapping the distribution atlas of NFATc1+PDGFRα+, NFATc1+PDGFRα-, and NFATc1-PDGFRα+ cells in adult homeostatic mice. In the next step, we plan to explore the differentiation trajectory of NFATc1+/PDGFRα+ cells during development using single-cell sequencing and other methods.

(3) It is recommended to add figure labels to Figure 1B to facilitate reader comprehension.

Thank you for your valuable suggestion to improve the quality of this manuscript. We have modified Figure 1B in the revised manuscript as follows:

(4) Why compare 3D images from tissue clearing with 3D reconstructions of confocal imaging after consecutive tissue slicing?

Thanks for your important and helpful comments to improve the quality of this manuscript, and we are sorry for the insufficient statement.

The original intention of comparing the two methods was to is to draw more credible conclusions from multiple perspectives, thereby minimizing the limitations inherent in the singular use of current advanced imaging techniques. Indeed, the description in the previous manuscript could lead to misunderstandings among readers. Therefore, in the revised manuscript, we have modified and integrated the content of Results 4 section into the Discussion section to eliminate unnecessary verbosity and potential confusion.

Page 18 in the revised manuscript, “TC-based advanced imaging procedure can clearly visualize its 3D structure, reconstruct the whole across latitudes, and understand the spatial position and expression of each structure, which could avoid the bias of traditional single-layer slicing may cause, and provides a more intuitive and objective description of the existing situation. However, our results demonstrated TC still has some limitations…”

Page 19 in the revised manuscript, “The 3D sections reconstruction results, however, effectively addressed the issue of weak tdTomato signal and provide a clearer visualization of the distribution of Zsgreen and tdTomato signals. For example, the td-tomato signal in the root pump, which was almost completely unobservable by TC-based imaging, can be clearly seen using confocal imaging and 3D reconstruction (Figure 3C-D, Figure 6C-D, and Figure S4, Figure S12). However, compared to TC, the quality of 3D reconstruction of sections still relies on the angle and quality of the sections, with the section angle having a significant impact on the reconstruction outcome. In addition, because the slice itself has a certain thickness (10 μM in this study), which leads to the appearance of discontinuous in the final reconstructed image, and the aesthetics and accuracy could be affected to a certain extent. Also, unavoidable tissue damage during the sectioning process may result in the loss of some information. Therefore, a variety of different information could be obtained through two different imaging technologies, which prompt us to use the advanced experimental procedure according to the actual purpose.”

(5) The experimental results section does not specify the age of the mice used, which lacks clarity for the reader and makes it difficult to determine at what developmental stage the observed distribution of NFATc1+/PDGFR-α+ cells occurs.

Thank you for your suggestion. I apologize for overlooking this point. I only displayed the age of the mice in some of the figures. All the transgenic mice discussed in this article are adults around 12-14 weeks. I have added the specific weeks of age in the main text.

(6) What is the rationale behind selecting day 1, day 3, and day 5 as the experimental time points in Figure 2B?

Thanks for your questions. 48 hours after injection, TAM can be metabolized in the body and converted into 4-OHT, which then distributes thoroughly to various tissue systems through the bloodstream. Therefore, we chose to administer a booster dose 48 hours after the initial injection to ensure timely replenishment and achieve high labeling efficiency. This drug administration scheme has already been validated for feasibility in our preliminary studies.

(7) In Figure 2E, why is there a large area of red signal visible in the tooth enamel?

Thanks for your valuable comments and advice on how to improve the quality of this research article and our future work. As we discussed in the main text, the existing TC-based imaging techniques cannot meet the requirements for capturing as conspicuous tdTomato signals as ZsGreen, which may due to: (1) the editing efficiency of the DNA recombinase-mediated lineage-tracing system has limitations; (2) the lower presence of NFATc1+ cells in the region-of-interest (ROI) ensures weak signals of tdTomato; (3) the TC method as described may result in poor penetration of td-tomato fluorescence signals. Therefore, to clearly display the NFATc1+ cells in the ROI (periodontal ligament, pulp, and alveolar bone) as much as possible, we increased the intensity of excitation fluorescence of 561-channel of the Lightsheet fluorescence microscopy, which led to a large area of unrelated red signal in non-target areas (tooth enamel). In future work, we will further improve the TC procedure to shorten the sample processing time, and developing other transgenic mice to address this issue. Thanks again.

(8) In the text at Line 249, the author notes that PDGFRα+ cells are widely distributed, and NFATc1+ cells are primarily located in the pulp horns. What is the relevance of their distribution to their function?

Thank you very much for your suggestion. We found that PDGFRα+ cells are widely distributed in dental pulp tissue. Combined with the results from subsequent cell ablation experiments, it revealed that PDGFRα+ cells contribute to the formation of the odontoblast layer and the pulp core. In our supplementary data, we discovered through immunofluorescence staining that double-positive cells co-expressed AlphaV in the dental pulp, indicating that they possessed MSC components. We need to further investigate the relationship between their distribution and function in the future.

(9) In Line 301 of the text, there is a mislabeling of Figure 4. Please verify this carefully throughout the document.

Thank you for your efforts, and we are sorry for our negligence. We have made the necessary corrections and have meticulously reviewed the entire manuscript to ensure that there were no similar mistakes. The corresponding descriptions have been amended as below:

Page 11 in the revised manuscript, “After 11 days tracing, the number of PDGFR-α+ & NFATc1+ cells and PDGFR-α+NFATc1+ cells increased significantly (Figure 7)…”

(10) Between Lines 323 to 325, the author states: "the wider range of PDGFR-α+ cells than NFATc1+ cells were observed, which laid the foundation for our conjecture that NFATc1+ cells may contribute as subpopulation of PDGFR-α+ cells." This statement is inaccurate.

Thank you for your suggestions. We apologize for the inaccuracies in our description and have made corrections in the original text.

Page 12 in the revised manuscript, “the wider range of PDGFR-α+ cells than NFATc1+ cells were observed, we speculate that there may be a hierarchical relationship between the two.”

(11) The author is advised to combine the use of single-cell sequencing data for cell trajectory analysis to corroborate the differentiation relationships between NFATc1+/PDGFR-α+ cells, discussing their specific origins and final differentiation fates.

Thank you for your suggestion; it is very meaningful to us and will be the focus of our future research work.

(12) In the Results 4 section, the comparison between tissue clearing imaging and 3D reconstruction of consecutive tissue slices could be discussed in the discussion section.

We greatly appreciate your suggestions. We have revised and integrated the entire Results 4 section into the Discussion section to avoid unnecessary redundancy and misunderstandings. This adjustment also emphasizes that the goal of using two imaging techniques is to draw more credible conclusions from multiple perspectives, thereby mitigating the shortcomings of relying solely on existing advanced imaging methods. The revised content are as follows:

Page 18 in the revised manuscript, “TC-based advanced imaging procedure can clearly visualize its 3D structure, reconstruct the whole across latitudes, and understand the spatial position and expression of each structure, which could avoid the bias of traditional single-layer slicing may cause, and provides a more intuitive and objective description of the existing situation. However, our results demonstrated TC still has some limitations…”

Page 19 in the revised manuscript, “The 3D sections reconstruction results, however, effectively addressed the issue of weak tdTomato signal and provide a clearer visualization of the distribution of Zsgreen and tdTomato signals. For example, the td-tomato signal in the root pump, which was almost completely unobservable by TC-based imaging, can be clearly seen using confocal imaging and 3D reconstruction (Figure 3C-D, Figure 6C-D, and Figure S4, Figure S12). However, compared to TC, the quality of 3D reconstruction of sections still relies on the angle and quality of the sections, with the section angle having a significant impact on the reconstruction outcome. In addition, because the slice itself has a certain thickness (10 μM in this study), which leads to the appearance of discontinuous in the final reconstructed image, and the aesthetics and accuracy could be affected to a certain extent. Also, unavoidable tissue damage during the sectioning process may result in the loss of some information. Therefore, a variety of different information could be obtained through two different imaging technologies, which prompt us to use the advanced experimental procedure according to the actual purpose.”

(13) The article only demonstrates the impact of removing PDGFR-α+ cells on the dental pulp and periodontal tissues of adult mice. What would be the impact of removing NFATc1α cells on teeth and periodontal tissues?

Thank you for your suggestions. Our lab had been investigating the role of NFATc1+ cells in PDL and dental pulp tissues which is currently submitted to another journal. So please forgive me for not being able to present the data. The ablation assays showed that NFATc1+ cells may be involved in the formation of the odontoblast layer in dental pulp and in promoting osteogenic differentiation in the periodontal ligament.

(14) The effects of removing PDGFR-α+ cells on the teeth and periodontal tissues of adult mice are shown in the article. What would be the impact on teeth and periodontal tissues if PDGFR-α cells were removed during early development?

Thank you for your question. Our current research has not yet focused on the impact of PDGFR-α+ cells on the formation of periodontal ligaments and dental pulp tissue during the developmental stage. In our literature search, we found articles indicating that PDGFR-α was expressed at all stages of tooth development, and that PDGFR-α signaling was crucial for regulating the growth of the tooth apex and the proper extension of the palatal shelves during palatal fusion. Disruption of PDGFRα signaling interferes with apex growth and the critical extension of palatal shelves during craniofacial development. In the future, we would like to focus on the role of PDGFR-α cells during teeth development.

(15) If the data on the skull are not presented in this paper, it is suggested not to overly describe it in the results section, or to include related skull data in supplementary figures.

We appreciate your attention to detail and your suggestions for improving the clarity and presentation of our work. The corresponding results of cranium and cranial sutures region were shown in Video S7-9 in the revised manuscript.

**Reviewer #3 (Recommendations For The Authors):**

We sincerely appreciate your thorough review and positive feedback on our manuscript. In accordance with your recommendations, all the questions and concerns have been well-addressed in the revised manuscript. We believe these revisions further enhance the clarity and quality of our work. The point-to-point reply to the comments is listed below:

(1) In line 181, the author claimed that "we modified and improved a rapid and efficient procedure...this ultrafast clearing technique could minimize the impact on transgenic mice." However, there is no mention in the main text of the amount of time required for other methods. How can the "rapid" element of your improved method be reflected? The author should briefly list a few other studies and discuss them.

Thanks for your important and helpful comments, and we are sorry for the insufficient statement. In recent years, a variety of tissue clearing methods have emerged. Here is a summary of the methods and durations used for hard tissue clearing as published in several authoritative journals:

**Author response table 1. sa2table1:** 

Methods	Durations	References
MarShie tissueclearing procedure	13-15 days	Nat Commun. 2024, 15, 1764
FDISCO clearing procedure	4.5 days	Sci Adv. 2019, 5, eaau8355.
PEGASOS method	12 days	Cell Research. 2018, 28, 803-818.
iDISCO	4.5 days	J Comp Neurol. 2024 532, e25582.

In comparison, our approach requires only approximately two days, thereby minimizing the potential damage to the tissue itself. Additionally, the study employs transgenic mice mediated by lineage tracing, and the shorter processing time also serves to reduce the impact on the fluorescence of the positive cells to a minimum.

(2) In Figure S6, the author mentioned the use of another 3D reconstruction method-DICOM-3D. What is the advantage of this methodology? Is the conclusion drawn the same as the previous approaches? The author should propose corresponding discussions in this section.

We sincerely appreciate your comments. The purpose of employing DICOM-3D reconstruction for the serial section images is to validate the constructed results obtained by Imaris. This method is based on sequential 2D DICOM images and utilizes 3D reconstruction and visualization technology to generate a stereoscopic 3D image with intuitive effects. Compared to Imaris reconstruction, this method offers a more straightforward and time-efficient approach. Regardless of the different reconstruction methods employed in this study, the ultimate goal remains consistent, which is to jointly address the spatial positioning and hierarchical relationships of PDGFR-α+, NFATc1+, and PDGFR-α+NFATc1+ cells from multiple perspectives, to enhance the credibility and persuasiveness of our results. We have also included the corresponding description in the revised manuscript as follows:

Page 8-9 in the revised manuscript, “To enhance the comprehensive and accurate display of the reconstruction results and to mitigate the potential errors that may arise from relying on single reconstruction method, we employed an alternative 3D reconstruction method—DICOM-3D. This method is based on sequential 2D DICOM images and utilizes 3D reconstruction and visualization technology to generate a stereoscopic 3D image with intuitive effects, which was a comparatively straightforward and highly efficient approach. We transformed the serial IF images into DICOM format and subsequently reconstruct it, and the same conclusion can be drawn, namely, PDGFR-α+ cells almost constituted the whole structure of pulp and PDL, with NFATc1+ cells as subpopulation (Figure S6).

(3) Line 292: Why was the tdTomato signal in confocal-based reconstruction more conspicuous than the TC procedure? Some descriptions would be beneficial for readers' understanding.

Thank you very much for your comments. We hypothesize that the current light-sheet systems have inherent limitations in capturing tdTomato signals of intact tissue, which become more evident in tissues with inherently low fluorescence strengths (in this work, due to the limitations of editing efficiency in DNA recombinase mediated lineage-tracing system, which guaranteed weaker tdTomato signal compared to ZsGreen). In contrast, traditional confocal imaging techniques do not encounter such issues. The corresponding descriptions in the revised manuscript are shown as follows:

Page 11 in the revised manuscript, “We hypothesize that the current light-sheet systems for intact tissue-imaging have inherent limitations in capturing tdTomato signals, which become more evident in tissues with inherently low fluorescence strengths (in this work, due to the limitations of editing efficiency in DNA recombinase mediated lineage-tracing system, which guaranteed weaker tdTomato signal compared to ZsGreen). In contrast, traditional confocal imaging techniques do not encounter such issues.”

(4) Part 2.2, line 305: What is the purpose of analyzing the cranium and cranial sutures region through TC technology?

Thank you for your comments. There are three main purposes of this part of the experiment. First, our research group has long been committed to studying the distribution and role of NFATc1+ SSCs in a variety of hard tissues, and our previous study has identified the presence of NFATc1+ cells in the cranium by single-cell sequencing. Therefore, in this work, we also intend to investigated the spatiotemporal atlas of NFATc1+ and PDGFR-α+ cells in cranium and cranial sutures region based on transgenic lineage tracing techniques. Second, as a part of craniomaxillofacial hard tissue, we intended to explore whether the presence of NFATc1+ and PDGFR-α+ cells in cranial bone tissue/suture is different from dental and periodontal tissue; In addition, the results in Video S7-9 further demonstrated that our improved tissue clearing procedure in this work is universal for a variety of hard tissues, which lay a foundation for our future researches.

Page 11 in the revised manuscript, “As a part of craniomaxillofacial hard tissue, we also intended to explore whether the presence of NFATc1+ and PDGFR-α+ cells in cranial bone tissue/suture is different from dental and periodontal tissue (our previous study has identified the presence of NFATc1+ cells in the cranium by single-cell sequencing28”)

(5) Some images before & after the tissue-clearing procedure need to be provided in the supplemental file.

Thanks for your important and helpful comments to improve the quality of this manuscript. We have included the corresponding description and photographs in the main text and the supplemental file as follows:

Page 7 in the revised manuscript, “As shown in Figure S1A-B, we recorded bright-field images of the maxilla before and after clearing, and our procedure achieved high transparency of the whole tissue. On this basis, whole-tissue imaging can be achieved, with the observation of different cell type distribution in spatial 3D structure.”

(6) In part 5, line 394, the author investigated the consequences of the ablation of PDGFR-α+ cells in dental pulp and periodontal mesenchymal tissues, but some research objectives and mechanisms need to be discussed here, regarding: "why choosing to ablation PDGFR-α+ cells instead of NFATc1+ cells? Was the hierarchical relationship between PDGFR-α+ cells and NFATc1+ cells considered during the experimental design?", etc.

Thank you very much for your suggestion, it has been very helpful. We chose PDGFR-α+ cells as the subject for the cell ablation experiments based on the results from the previous lineage tracing and hierarchical relationship studies. We have included the corresponding description and photographs in the main text and the supplemental file as follows:

Page 13 in the revised manuscript, “The results from the aforementioned lineage tracing experiments showed that PDGFR-α+ cells constitute a significant component of both dental pulp and periodontal tissues. Additionally, the hierarchical relationship experiments revealed that a portion of NFATc1+ cells in the periodontal ligament derives from PDGFR-α+ progenitor cells. Therefore, investigating the role of PDGFRα+ cells in dental pulp and periodontal tissues has become more urgent.”

(7) Some claims in the main text were lack of literature citation, such as in lines 207 and 234.

Thank you very much for your comments. We are deeply sorry for the mistakes. We have added the relevant references at the appropriate locations in the main text as follows:

(1) line 207 of previous manuscript (page 8, line 206 in the revised manuscript): We sincerely apologize for the typo that occurred during the writing process and have revised the original text to: which was consistent with RNA-sequencing results in the previous study20. (DOI: 10.1177/00220345221074356)

(2) line 234 of previous manuscript (page 9, line 234 in the revised manuscript): “we employed an alternative 3D reconstruction method—DICOM-3D27.” (DOI: 10.1177/09544119211020148)

(8) What were the specific reasons for the conspicuous tdTomato signal in the reconstructed images obtained by traditional serial section-based confocal imaging, which were not as evident in TC imaging?

Thank you very much for your comments. Traditional sectioning and subsequent confocal imaging can clearly display fluorescence signals on a single plane (Figure 3B, Figure 6B, Figure S3, S8, S11, S16, S19), therefore, after 3D reconstruction of multiple planes, it will still have a high resolution (Figure 3, 4, 7, 8). However, for TC imaging, the current light-sheet systems have inherent limitations in capturing tdTomato signals of intact tissue, which become more evident in tissues with inherently low fluorescence strengths (in this work, due to the limitations of editing efficiency in DNA recombinase mediated lineage-tracing system, which guaranteed weaker tdTomato signal compared to ZsGreen). In contrast, traditional confocal imaging techniques do not encounter such issues.

(9) In tissue clearing techniques, do the chemical reagents and procedures used affect the signal intensity of tdTomato and Zsgreen?

We appreciate your helpful comment. In this work, we modified and improved a rapid and efficient tissue deep clearing (TC) procedure based the existing SUMIC method, and (Nature Cardiovascular Research, 2024, 3, 474–491; Cell, 2023, 186, 382-397.e24.). These researches have confirmed that the chemical reagents used in this method do not affect the inherent fluorescence signal of transgenic animals. With our improvements, we minimized the sample processing time as much as possible to avoid any potential adverse effects. The results in Figure 2, Figure 5, and Figure S1 indicated that after TC procedure, the tissue exhibit significant ZsGreen signals and certain tdTomato signals, which sufficiently support our conclusions.

(10) How did you address the issue of sample integrity and discontinuities in the z-axis caused by the stratification of slices in your reconstructions?

We greatly appreciate your comments. Currently, reconstruction techniques based on continuous sectioning cannot fully eliminate the discontinuities in the z-axis. Therefore, it is for this reason that we need to compensate for this deficiency by imaging the whole tissue through TC procedure. These two 3D-reconstruction and imaging technologies complement each other to jointly address the spatial positioning and hierarchical relationships of PDGFR-α+, NFATc1+, and PDGFR-α+NFATc1+ cells from multiple perspectives. Additionally, this deficiency can be minimized by improving the technical skills, reducing section thickness, and to minimize tissue loss during sectioning, which is our future research endeavors.

(11) In Figure 2B, the schematic representation of the operational principle "Cre-loxp/Dre-loxp" does not correspond to the genotype "CreER/DreER". Please correct it.

Thanks for your important comments. We are sincerely sorry for the mistake. We have modified Figure 2B in the revised manuscript as below:

(12) Line 450, the specific distribution and differences of PDGFR-α+, NFATc1+, and PDGFR-α+&NFATc1+ cells in pulp and periodontal tissues need to be further described and explained.

Thank you for your question. We have described this part on page 16 in the revised manuscript, “In PDL tissue, pulse data demonstrated widespread and abundant expression of PDGFR-α single-positive cells as well as NFATc1 single-positive cells, with no significant alteration in expression pattern or quantity after lineage tracing. Consequently, we conclude that in periodontal ligament and dental pulp tissues, PDGFR-α single-positive and NFATc1 single-positive cells primarily label intrinsic periodontal mesenchyme in PDL. Conversely, PDGFR-α+&NFATc1+ cells exhibited a more confined localization in PDL. The tracing data clearly illustrated that PDGFR-α+&NFATc1+ cells successfully gave rise to numerous progenies, which become predominant constituents within the periodontal ligament. In pulp tissue, the distribution of PDGFR-α single-positive cells was similar as that in PDL, primarily labeled odontoblast cell layer and there was not a significant increase in ZsGreen signal after tracing assay.”

(13) In Figure S9, the sparse presence of NFATc1+ cells in pulp and periodontal tissue raises questions about the plasticity and differentiation potential of these cells. The author should include relevant discussions in this section.

Thanks for your suggestion. Considering the plasticity and differentiation potential of NFATc1+ cells, we conducted immunofluorescence staining and found that the PDGFR-α+&NFATc1+ cell lineage in dental pulp and periodontal tissues represents a heterogeneous population. This population includes non-terminally differentiated mesenchymal stem cells (MSCs) as well as hematopoietic cells, indicating significant heterogeneity. We have also added this part of the discussion on page 17 of the manuscript.

Page 17 in the revised manuscript, “Cell ablation and immunofluorescence staining experiments further characterized the types and functions of PDGFR-α+/PDGFR-α+&NFATc1+ populations. After ablating PDGFR-α+ cells, we observed damage to the odontoblast layer and shrinkage of the pulp core in dental pulp tissue, indicating that PDGFR-α+ cells contribute to the composition of dental pulp tissue, particularly the odontoblast layer (Figure. 9C, D). In the periodontal ligament, we noted a reduction and destruction of collagen fibers, suggesting a role for PDGFR-α+ cells in periodontal tissue structure (Figure. 9E, F). Previous results confirmed the presence of double-positive cells in both dental pulp and periodontal tissues and provided insights into their hierarchical relationships in the periodontal ligament (Figure. 8). To further investigate the double-positive cell population, we developed an inducible dual-editing enzyme reporter system to label these cells with tdTomato signals. Using AlphaV as a marker for mesenchymal stem cells (MSCs) and CD45 for hematopoietic cells, we found that double-positive cells included components of both MSCs and hematopoietic cells (Figure S22B, C), indicating a heterogeneous population. Further experiments are necessary to determine whether the predominant role in this co-positive MSC population is played by PDGFR-α+ or NFATc1+ and to clarify the specific functions of these cells in the future.”

(14) Part 3, line 351, the authors were unable to confirm the hierarchical relationship between PDGFR-α+ and NFATc1+ cells in the dental pulp region. Could this be due to limitations in experimental design or technical methods? Have you considered other factors that might explain these results?

Thank you for your question. We believe that the possible reason was that PDGFR-α+ cells were a widely distributed constitutive component of dental pulp tissue, while NFATc1+ cells had a more limited expression range, resulting in a significant difference between the two. Therefore, we were unable to calculate the differences. In the future, we could further investigate the hierarchical relationship between the two by increasing the sample size or through in vitro experiments such as immunoprecipitation.